# Unraveling Nitrogen Fixing Potential of Endophytic Diazotrophs of Different *Saccharum* Species for Sustainable Sugarcane Growth

**DOI:** 10.3390/ijms23116242

**Published:** 2022-06-02

**Authors:** Rajesh Kumar Singh, Pratiksha Singh, Anjney Sharma, Dao-Jun Guo, Sudhir K. Upadhyay, Qi-Qi Song, Krishan K. Verma, Dong-Ping Li, Mukesh Kumar Malviya, Xiu-Peng Song, Li-Tao Yang, Yang-Rui Li

**Affiliations:** 1Key Laboratory of Sugarcane Biotechnology and Genetic Improvement (Guangxi), Ministry of Agriculture, Sugarcane Research Center, Chinese Academy of Agricultural Sciences, Guangxi Key Laboratory of Sugarcane Genetic Improvement, Sugarcane Research Institute, Guangxi Academy of Agricultural Sciences, Nanning 530007, China; rajeshsingh999@gmail.com (R.K.S.); anjneysharma@gmail.com (A.S.); gdj0506@163.com (D.-J.G.); drvermakishan@gmail.com (K.K.V.); malviyamm1983@gmail.com (M.K.M.); 2Guangxi Key Laboratory of Crop Genetic Improvement and Biotechnology, Nanning 530007, China; xiupengsong@gxaas.net (X.-P.S.); liyr@gxu.edu.cn (L.-T.Y.); 3School of Marine Sciences and Biotechnology, Guangxi University for Nationalities, Nanning 530008, China; singh.pratiksha23@gmail.com; 4College of Agriculture, State Key Laboratory of Conservation and Utilization of Subtropical Agro-bio Resources, Guangxi University, Nanning 530005, China; 5Department of Environmental Science, V.B.S. Purvanchal University, Jaunpur 222003, India; sku.env.lko@gmail.com; 6Guangxi Subtropical Crop Research Institute, Sugarcane Research Institute, Nanning 530001, China; alice771992@126.com; 7Microbiology Institute, Guangxi Academy of Agricultural Sciences, Nanning 530007, China; lidongping0201@126.com

**Keywords:** endophytic diazotrophs, high-throughput sequencing, microbial diversity, nitrogen fixation, *nifH* gene, sugarcane

## Abstract

Sugarcane (*Saccharum officinarum* L.) is one of the world’s highly significant commercial crops. The amounts of synthetic nitrogen (N_2_) fertilizer required to grow the sugarcane plant at its initial growth stages are higher, which increases the production costs and adverse environmental consequences globally. To combat this issue, sustainable environmental and economic concerns among researchers are necessary. The endophytic diazotrophs can offer significant amounts of nitrogen to crops through the biological nitrogen fixation mediated *nif* gene. The *nifH* gene is the most extensively utilized molecular marker in nature for studying N_2_ fixing microbiomes. The present research intended to determine the existence of novel endophytic diazotrophs through culturable and unculturable bacterial communities (EDBCs). The EDBCs of different tissues (root, stem, and leaf) of five sugarcane cultivars (*Saccharum officinarum* L. cv. Badila, *S. barberi* Jesw.cv Pansahi, *S. robustum*, *S. spontaneum*, and *S. sinense* Roxb.cv Uba) were isolated and molecularly characterized to evaluate N_2_ fixation ability. The diversity of EDBCs was observed based on *nifH* gene Illumina MiSeq sequencing and a culturable approach. In this study, 319766 operational taxonomic units (OTUs) were identified from 15 samples. The minimum number of OTUs was recorded in leaf tissues of *S. robustum* and maximum reads in root tissues of *S. spontaneum.* These data were assessed to ascertain the structure, diversity, abundance, and relationship between the microbial community. A total of 40 bacterial families with 58 genera were detected in different sugarcane species. Bacterial communities exhibited substantially different alpha and beta diversity. In total, 16 out of 20 genera showed potent N_2_-fixation in sugarcane and other crops. According to principal component analysis (PCA) and hierarchical clustering (Bray–Curtis dis) evaluation of OTUs, bacterial microbiomes associated with root tissues differed significantly from stem and leaf tissues of sugarcane. Significant differences often were observed in EDBCs among the sugarcane tissues. We tracked and validated the plethora of individual phylum strains and assessed their nitrogenase activity with a culture-dependent technique. The current work illustrated the significant and novel results of many uncharted endophytic microbial communities in different tissues of sugarcane species, which provides an experimental system to evaluate the biological nitrogen fixation (BNF) mechanism in sugarcane. The novel endophytic microbial communities with N_2_-fixation ability play a remarkable and promising role in sustainable agriculture production.

## 1. Introduction

Sugarcane is an important crop has been cultivated for more than 2200 years [1] in tropical and subtropical countries and provides a source of crystal sugar, renewable energy, paper production, and raw material for many biomass-based products. It is a C_4_ perennial crop with four different phases of development such as germination, tillering, grand growth, and maturity, probably practiced with varying levels of chemical fertilizers to affect its growth and development during these phases. The utilization of chemical fertilizers such as nitrogen (Urea), in China ranged from 400 to 800 kg ha^−1^, which are considerably higher than rates utilized in other nations. For example, in Australia, Brazil, and India, the application levels of N were 60–100, 150–400, and 160–200 kg ha^−1^ year^−1^, respectively [2]. The excess utilization of chemically synthesized N_2_ fertilizer raises many challenges to environmental sustainability. Excess use of chemically synthesized N2 fertilizers in agricultural soil triggers denitrification, soil acidification, groundwater pollution, and greenhouse gas emissions [3,4,5]. According to FAO, China is the world’s largest producer and consumer of chemical N_2_ (Urea) fertilizer, accounting for around one-third of total global usage (FAOSTAT, www.fao.org (accessed on 15 February 2021)) [6]. From the accessible data in 2017, based on the Statistic Division of the Food and Agriculture Organization of the United Nations (FAOSTAT), China (29.62 million tonnes), United States of America (USA; 11.58 million tonnes), Indonesia (2.95 million tonnes), Canada (2.47 million tonnes), and Vietnam (1.69 million tonnes) are the five countries using the maximum amount of nitrogen-based fertilizers for agricultural production [7].

Only microorganisms found in the soil, rhizosphere, and plant tissues including cyanobacteria in fresh and seawater bodies are recognized to be capable of N_2_-fixation, termed as diazotrophs. Additionally, this response is completed with the presence of the nitrogenase enzyme, which is a complex of proteins encoded by genes *nifD*, *nifH*, and *nifK* [8]. Out of these genes, *nifH* gene has been most commonly employed to explore the diversity and conformation of diazotroph communities, because the *nifH* gene is highly conserved [9]. Moreover, *nifH* gene sequencing has revealed the diazotroph species that occur in the diverse group of microbiomes in the different crops and are a subdivision of the soil or plant diazotroph community [10]. Therefore, use of the *nifH* gene as a molecular marker has resulted in substantial breakthroughs in our knowledge regarding the ecology and function of diazotrophs. Previously, this *nifH* gene has been largely studied in our laboratory based on culture-dependent methods [11,12,13,14], but our growing interest is culture-independent to understand the various diazotrophs of microbial communities (MCs) that are found in tissues (leaf, stem, and root) of different sugarcane species. Furthermore, the diazotrophic group of *Saccharum* sp. might be different when it is grown in different types of soils as well as environmental conditions [15]. It is unknown which types of microbes may indicate the absence or presence of significantly fixing N_2_ and their activity in sugarcane. Therefore, complete knowledge and the mechanisms of diazotrophic bacteria that are applied to transfer the studies from bacterium to sugarcane or other crops through BNF remains a major, ongoing research area.

Unculturable and culturable diazotrophic bacterial endophytes (DBEs) that are useful to plants and fix N_2_ from different sugarcane species are the focus of this manuscript. These microbes mostly colonize root tissues, but they also colonize other plant components such as leaves, stems, nodules, fruits, and seeds [16,17]. Nitrogen is found in dinitrogen (N_2_) form, which is inert and cannot be utilized by plants. To be used by plants, this N_2_ must be reduced or fixed into forms such as nitrate (NO_3_^−^) and ammonium (NH_4_^+^). Bacterial endophytes live in plant tissues and may provide a favorable environment for N-fixation by reducing competition with other microorganisms in the rhizosphere and maybe providing a microaerobic condition required for nitrogenase activity [18,19].

There is a rising interest in understanding how the diazotrophic microbial communities of bacterial endophytes (BEs) found in sugarcane respond to biofertilizers such as fixed N_2_. Whereas, metagenomics approaches have considerably improved our understanding of how these DBEs of microbial communities are organized and how many different types of N_2_-fixing microbial species are present along with some new genera of species that exist. However, unfortunately, owing to the drawbacks of the culture-based approach, many microorganisms remain unidentified and uncultured. Therefore, the studies on DMCs using omics technologies offer an opportunity for improving the methods of isolation techniques to isolate the unknown diazotrophs found in different tissues of sugarcane varieties yet uncultured, which helped agricultural researchers. Here, we study the *nifH* gene using microbial diversity analysis, including the number of different endophytic bacterial species (richness) and their relative abundance (evenness) in the root, stem, and leaf of sugarcane microflora. Subsequently, microorganisms represent a majority of the diazotrophic biodiversity found in sugarcane plants, and their functions involved an N_2_-fixation using the metagenomics approach.

Our main goal was to evaluate the presence of endophytic diazotrophic bacterial communities (EDBCs) and to characterize the N_2_-fixing bacteria found in different sugarcane species using a cultivation-dependent and independent method. The primary problems of defining endophytic bacterial microbiota are that they are under-represented inside plant tissues due to their low cell population, limiting the depth and scope of bacterial community characterization. Further, the unculturable microbial diversity of diazotrophic species might be a great resource for determining novel concepts for sustainable agricultural practices and microbial applications to discover the diversity of diazotroph microbes in sugarcane and their possibly critical role in fertilizer. 

## 2. Results

### 2.1. Bacterial Diversity Using the Culture-Dependent Method

A total of 345 endophytic bacterial isolates were recovered from the various sugarcane plant components (root, stem, and leaf) using 10 different selective mediums (Figure 1A,B).

Out of them, only 194 isolates were selected for further study because these strains were morphologically different. These 194 strains were classified into 42 different bacterial species after 16S *rRNA* gene sequencing (Table 1). Additionally, of these all isolates were tested for nitrogenase activity, the results revealed a significant activity among all isolates that ranged from 6.163 ± 0.112 to 29.709 ± 0.538 nmoL C_2_H_4_ mg protein h^−1^, respectively. All isolates were positive for *nifH* gene amplification (Table 1).

### 2.2. Metagenomic Sequence Analysis of the nifH Gene

A total of 25 samples (five plants from each sample) were collected from five *Saccharum* species of sugarcane plants including *S. officinarum* L. cv. Badila, *S. barberi* Jesw. cv Pansahi, *S. robustum*, *S. spontaneum*, and *S. sinense* Roxb. cv Uba. We studied a total of 15 plant tissue samples for roots, stems, and leaves. Total genomic DNA was successfully extracted to establish the endophytic bacterial population profiles. PCR amplification was performed with the selected *nifH* gene primer. Roots, stems, and leaves with low DNA concentrations after PCR were discarded in preparation for sequencing.

To obtain higher quality and more accurate biological information about the microbial community, analysis of quality-filtered reads was undertaken after combining the *nifH* gene sequences from sugarcane that were extracted from the results. Appendix A provides the results obtained from the preliminary analysis of effective and optimized sequence length distribution. Table 2 provides a summary of statistics for high-quality reads to optimize sequence data used for subsequent clustering OTUs and species information analysis of total sequences were obtained. After optimizing the original sequence of the *nifH* gene, the optimized sequence length of all five sugarcane samples in different tissue regions of different samples and among sugarcane groups was found to, on average, range from 290 to 360. A total of 319,766, sequence reads were collected, and the minimum number of reads recorded to *S. robustum* in the leaf sample was 11,358, whereas the maximum reads were *S. spontaneum* with the root, for which 31,283 reads were obtained with a coverage value for all sugarcane plant tissue samples of 0.99. The number of sequenced reads for individual samples per plant compartment and assigned OTUs, as well as the average amount of reads before and after quality monitoring and trimming from sugarcane, are presented in Table 2. The raw data of *nifH* gene sequences obtained in this study were deposited in the NCBI Sequence Read Archive (SRA) database under the accession number PRJNA810252.

### 2.3. Richness and Evenness of Species

A rarefaction curve is a statistical approach used to evaluate species richness based on the sample results. This method is commonly used in OTUs and is extremely beneficial in environmental sample analysis. It can be utilized to assess if a specific sample has been sequenced sufficiently to represent its identity. This curve is applied to calculate species richness for a given number of samples and plot the number of species against the number of samples. The intra-sample richness and diversity of the observed OTUs indicated that the sequencing depth was sufficient to capture all the diversity presented. For the root, stem, and leaf samples, the curve represents that the sequencing effort was sufficient to obtain the most abundant bacterial OTUs. Among this group, UL was found to be high in OTU count and reasonable sequence number (Appendix A). The all-curve graph of each sample was completed at a 97% similar level. It may also be utilized to compare the abundance of species in each sample with varying quantities of sequencing data by using the method of random sampling of sequences. When the curves tended to be flat this indicated that the number of reads was significant enough to reflect the species richness.

### 2.4. Operational Taxonomic Units (OTU) Analysis

Based on the Venn chart, used to count the number of unique OTUs found in five sugarcane species, with successive increases in OTUs of the stem, the leaf moved further to the root (Figure 2). A total of 1322 OTUs were found and out of these 828 are unique OTUs found in all sugarcane *Saccharum* species. These unique sequences were found in three different parts of the plant, i.e., leaf, stem, and root. From the chart (Figure 2A), we can observe that the leaf and stem groups reported nearly similar unique numbers (256 and 255) but, significantly, the root showed more OTUs (317) than the other two groups at the 97% cut-off level. To better display the number of unique and shared OTUs in each sample of individual tissues, we display it in the form of a petal map (Figure 2B). 

The OTU analysis of different sugarcane varieties between the different tissues was performed through principal component analysis (PCA), to calculate the percentage of each OTU in each sample. The major principle for the assemblage of every tissue group to the individual sample was the relationship between different OTUs. A scatterplot diagram of the PCA showed that the OTUs of different sugarcane tissues accounted for 25.63% of variation in PC1 and 18.20% for PC2 in the selected sugarcane varieties (Figure 2C). Points of various colors or forms indicate sample groups in various habitats or conditions. PC1 and PC2 represent the suspected influencing factors for the microbial composition of the two groups.

Figure 2D is a species accumulation curve that was utilized to describe the increase in species sampled, and it is an effective resource for understanding the species composition of surveyed plots and predicting species richness (SR). It is frequently used in biodiversity and community surveys to determine the competence of the sample quantity and estimate the SR based on the curve’s properties. If the curve rises rapidly, nearly in a straight line, it demonstrates that the sample quantity is insufficient and must be raised; if the curve slows down after the fast rise, sampling is adequate.

A total of 40 bacterial families and 58 genera with others and the unclassified group were detected. The *nifH* microbial communities of relative abundances were varied among all five sugarcane species in different tissues compartment studied. The most abundant *nifH* bacteria were identified at the genus level are *Saccharum, Ideonella, Kosakonia, Bradyrhizobium, Burkholderia, Azospirillum, Herbaspirillum, Rhizobium, Sorghum, Pseudacidovorax, Enterobacter, Stenotrophomonas, Methylobacterium, Desulfocapsa, Klebsiella, Dechloromonas, Ruficoccus, Rhodoplanes, Ampullimonas, Desulfovibrio, Paenibacillus, Geobacter, Vitreoscilla, Scytonema, Sulfuricurvum, Magnetospirillum, Sphingomonas, Rhodopseudomonas, Xanthobacter, Methyloversatilis, Panicum, Corallococcus, Lachnoclostridium, Azoarcus, Azohydromonas, Thermincola, Paraburkholderia, Zea, Tolumonas, Rhodococcus, Pleomorphomonas, Rubrivivax, Beijerinckia, Aegilops, Ensifer, Spinacia, Phleum, Calenema, Azonexus, Hymenobacter, Azorhizobium, Pseudodesulfovibrio, Sinorhizobium, Escherichia, Bacillus,* and *Exophiala.*

### 2.5. Metagenome Analysis Revealed Insight into the Major Microbial Taxonomic Classification from Sugarcane

In Figure 3, the complete taxonomic classification from Kingdom to Genus level is also represented as Krona graphs. A total of 87,023 (leaf), 99,485 (stem), and 133,258 (root) non-chimeric sequence reads from different sugarcane plants were used (OTU selected) to display the Krona analysis. It looks like a pie chart, in that it splits a separate class into segments, but with an embedded hierarchy, i.e., each segment is covered with minor areas. In the *nifH* gene of endophytic microbial communities, Proteobacterium, Streptophyta, Actinobacteria, Cyanobacteria, Firmicutes, and Verrucomicrobia were found to be dominating phyla across all the tissues of sugarcane samples. In the root sample *nifH* gene, the proteobacteria was the most dominant phylum (Figure 3). *Saccharum*, Proteobacteria, and others all were the most dominant phyla, accounting for approximately 49, 46, and 15 percent of the total sequences from sugarcane leaf tissues, respectively (Figure 3A). The microbial taxa detected in the stem of endophytic microbial *nifH* community species are different from the leaf in all different sugarcane species. *Saccharum* (20%), Proteobacteria (70%), and all others (10%) are more commonly found in the stem, but at the genus level, Proteobacteria are found more in the *nifH* gene producing microbes such as *Herbaspirillum, Klebsiella, Magnetospirillum, Pseudacidovorax, Xanthobacter, Azospirillum, Burkholderia, Kosakonia, Rhizobium, Brevibacterium, Paenibacillus, Achromobacter, Sinorhizobium, Desulfovibrio, Xanthobacter, Bradyrhizobium,* and *Sphingomonas* (Figure 3B). In the root at the phylum level, Proteobacteria and Actinobacteria were found to be the most occupied phyla among all tissues (leaf and stem) samples of five sugarcane species accounting for 99% of the population in *Azospirillum* (11%), *Herbaspirillum* (11%), *Burkholderia* (17%), *Pseudacidovorax* (7%), *Ideonella* (8%), *Stenotrophomonas* (5%), *Bradyrhizobium* (3%), and *Methylobacterium* (3%). Except for Proteobacteria, each of the minor phyla were represented by a small number of bacterial sequences, only 0.5 and 0.2 % (Figure 3C).

### 2.6. Star Map Analysis of 10 Abundance Genus 

Figure 4 shows, at the genus level classification, a star map for the species abundance of the top 10 most abundant species present in different sugarcane tissues of sugarcane varieties. This map represents the relative abundance of a sample. The fan shape in each star chart represents a species, which is distinguished by different colors. The radius of the fan is used to represent the relative abundance of the species. The longer and smaller the radius of the fan is, the higher and lower the abundance of microbial species present in sugarcane.

### 2.7. Diversity Analysis of nifH Gene Sequences in Five Sugarcane Species

#### 2.7.1. α-Diversity of Bacterial Community

In this study, the statistical difference in alpha diversity revealed a similar level of OTU richness, and the diversity of endophytic diazotrophic microbial communities was determined. Table 3 displays the ace, Chao, Shannon, Simpson, sobs, and PD whole tree index suggesting that the sequencing depth is adequate to accurately describe the N-fixing bacterial microbial communities with sugarcane samples. Significant differences were observed between all alpha diversity indices of leaf, stem, and root among the 15 samples of five different sugarcane species (*p* < 0.03).

We observed that the number of nitrogen-fixing microbial species was significantly higher in the *S. Sinense* Roxb. cv Uba than in the others such as *S. Barberi* Jesw. cv Pansahi > *S. robustum*
*> S. spontaneum > S. officinarum* L. cv. Badila. Additionally, the range of the chao alpha diversity index was 429.65–283.00 of *S. sinense* Roxb. cv Uba, significantly higher than the lower of *S. officinarum* L. cv. Badila which ranged from 198.11 to 155.12 (Table 3). In the Shannon index, values of the *nifH* gene were also significantly higher in the *S. sinense* Roxb. cv Uba (range: 4.92–3.07) than those in other sugarcane species and lower values were found in *S. robustum* (range: 3.79–1.62). The other diversity indexes such as Simpson, sobs, and PD whole tree values also showed a similar trend and maximum in *S. sinense* Roxb. cv Uba except *S. robustum* in the Fimpson index (Table 3). 

#### 2.7.2. β-Diversity of Bacterial Community

Beta diversity analysis is applied to compare the difference in endophytic bacterial species among the samples collected from different sugarcane species. A variety of indexes can measure β-diversity but to better understand this resemblance we measured the generally used factors such as Bray–Curtis distance matrices, (Figure 5). Heat maps are ideal for visualizing large numbers of OTUs and can be utilized to identify classes of rows with similar values which are shown as areas with similar color boxes (Figure 5A). Additionally, the result of clustering is presented as the similarity values of OTUs depending on the distance metric. A dendrogram of the hierarchy is displayed as the result of hierarchical clustering on a heat map. As a result of the clustering calculation, the rows in the heat map all have the root (SR, BR, PR, RR, and UR) samples of different sugarcane species placed in the same cluster. Whereas, leaf and stem are placed together in another cluster, and this cluster was divided into three sub-groups. The first group has three samples (PL, BL, and BS), the second group contains only two samples of the stem (RS and US), and the third group contains five samples (SL, SS, PS, RL, and UL) (Figure 5A). The separation of microbiota structures across distinct branches was visualized using nonmetric multidimensional scaling (NMDS) plots based on the Bray–Curtis method. NMDS results showed the same trends between the *nifH* communities in roots of all tissue of the sugarcane variety, although some samples showed differences between different tissues of sugarcane in one group, such as leaf and stem. For diversity, we used principal coordinates analysis (PCoA) on Bray–Curtis to estimate the core components causing differences between the samples. Bray–Curtis PCoA showed variations between different tissues (leaf, stem, and roots) of five sugarcane species with the first principal (PC1) and the second principal (PC2) axis explaining 18.90 and 9.37% of the total variation. PCoA of the method was utilized to construct an evaluation of the phylogenetic tree differences between samples. The endophytic bacterial community in all sugarcane plant tissues is clustered in two main branches on the hierarchical clustering tree. The *nifH* MCs in the different sugarcane species were clustered indicating a clear separation between the communities in root samples and nearly all belonged to one cluster (Figure 5B).

### 2.8. Community Composition Histogram and Phylogeny Tree of Diazotrophs

The microbial communities of relative abundances varied among the five sugarcane species based on the genus and species level of annotation and statistically studied in each sample, shown in Figure 6. The sugarcane endophytic root diazotroph community was distinct from those of stem and leaf tissue samples in both levels of classification. All sample sequences were classified into a total of 56 different genera including *Saccharum, Ideonella, Kosakonia, Bradyrhizobium, Burkholderia, Azospirillum, Herbaspirillum, Rhizobium, Sorghum, Pseudacidovorax, Enterobacter, Stenotrophomonas, Methylobacterium, Desuilfocapsa, Klebsiella, Dechloromonas, Ruficoccus, Rhodoplanes, Ampullimonas, Desulflovibrio, Paenibacillus, Geobacter, Vitreoscilla, Scytonema, Sulfuricurvum, Magnetospirillum, Sphingomonas, Rhodopseudomonas, Xanthobacter, Methylovarsatilis, Panicum, Corallococcus, Lachnoclostridium, Azoarcus, Azohydromonas, Thermincola, Paraburkholeria, Zea, Tolumonas, Rhodococcus, Pleomorphomonas, Rubrivivax, Beijerinckia, Aegilops, Ensifer, Spinacia, Phleum, Calenema, Azonexus, Hymenobacter, Azorrhizobium, Pseudo desulfovibrio, Sinorrhizobium, Escherichia, Bacillus, Exophiala,* unclassified and others (<0.5%). The most abundant diazotrophic bacteria in the root samples of all sugarcane species belonged to the same hierarchical clustering group based on community composition between samples (Figure 6A). Whereas, in RS, sample *Kosakonia* was the most abundant bacterial genus and present in a single cluster. However, the other nine samples were divided into two groups, and one group contains seven (SS, RL, UL, SL, PS, PL, and BL) samples, while the other has only two (US and BS) samples (Figure 6A). The top eight genera that were present to fix nitrogen fixation in the tested sugarcane samples are *Kosakonia, Bradyrhizobium, Burkholderia, Azospirillum, Herbaspirillum, Rhizobium, Enterobacter,* and *Stenotrophomonas.* In the diazotroph communities based on species-level classification, there were 94 different species in unclassified and others (<0.5%), as shown in Figure 6B. The all-sugarcane samples of leaf and stem were much less diverse than those of the root samples. The leaf and stem of sugarcane contained common plant-associated diazotroph genera and species, present in the same hierarchical clustering group based on community composition.

### 2.9. Analysis of Species Differences in Each Classification Level

Matastats analysis was used, showing the relative abundance of the diazotrophic microbial community of the top 20 at genus level in different species of sugarcane plants in different tissue groups, i.e., A (green-colored bars), B (green-colored bars), and C (blue-colored bars) (Figure 7). Only top 20 species with the highest abundance and differences were selected for line graph drawing if there were more than 20 different species. This statistics analysis in tissue-wise groups showed that the abundance of *Saccharum, Bradyrhizobium,* and *Burkholderia,* was significantly higher. The findings are provided in Figure 7, those with a *p*-value of <0.05 (significant difference) were chosen and presented in a line graph.

### 2.10. Analysis of Similarity (ANOSIM) Test

The Analysis of Similarity (ANOSIM) test was used to conduct the statistical testing of variance for microbial community composition changes between root, leaf, and stem of different sugarcane species (Figure 8). Similarities between the diazotrophic endophytic bacterial communities of different sugarcane species were compared by ANOSIM based on Bray–Curtis distance. *p*-values less than 0.001 were considered the significant differences. Additionally, ANOSIM obtained a significant change in the root, stem, and leaf microbiome composition among different sugarcane species (Figure 8A–C; R= 0.637, 0.536, and 0.734; *p* = 0.001). Based on the results of this analysis, all sugarcane species had most significantly changed root diazotrophic microbial communities.

### 2.11. Linear Discriminant Analysis Effect Size (LEfSe)

We used a linear discriminant analysis effect size (LEfSe) biomarker for nifH gene detection in the data obtained from different tissues of sugarcane plants. LEfSe analysis LDA results presented in Table 1 were udnertaken to assess if statistically significant variations in taxon abundance of distinct sugarcane species occurred, as well as to investigate the biological significance of the species in each treatment (Figure 9). LEfSe initially robustly identifies traits that are statistically distinct between biological classes. In the particular root, there were many taxa of different tissues with substantially different abundances in the five sugarcane species. The LEfSe study revealed that there were 38 potential bacterial clades (leaf—6; stem—4; and root—28) demonstrating statistically significant differences with LDA and biologically consistent variations in all tissues of different sugarcane plants. The rings in the cladogram from inside to outside show phylum to genus diazotrophic taxonomic levels. In these tests, biomarkers emerging for the leaf included Clostridiales, Clostridia, Poales, Liliopsida, Streptophyta, and Viridiplantae. Additionally, in the root, Rhizobiaceae, Rhizobiales, Puniceicoccaceae, and Puniceicoccales. In the root, biomarkers were mostly clustered in Proteobacteria, including Beijernickiaceae, Bradyrhizobiaceae, Methylobacteriaceae, Rhodospirilliaceae, Burkholderiaceae, Desulfovibrionaceae, Xanthomonadaceae, etc. (Figure 9).

## 3. Discussion

Sugarcane is a tropical and perennial crop that lacks the nodulating symbiotic diazotrophs. Therefore, these conclusions offer a foundation for future research, to study and identify free-living or associative novel diazotrophic microbes that contribute to this industrially significant crop for adaptability and growth improvement. In this investigation, we exclusively assessed the endophytic diazotrophic microbes found in different tissues of sugarcane plants. Previously, we observed in sugarcane field samples, that significant shifts were observed in diazotrophic bacterial community composition between different sugarcane varieties and their plant compartments, i.e., root, stem, and leaf. We provide a custom-designed method for better analysis of N_2_-fixing MCs utilizing Illumina’s next-generation sequencing technology in this paper. In our new approach, we showed that the root compartment is a hot spot for well-known diazotrophs, and it contains the highest diazotroph diversity as compared to the leaf and stem compartments of the sugarcane plant. Because of their capacity to absorb nutrients from the soil, fix atmospheric N_2_, promote nutrient solubilization, and function as biocontrol agents, microbial community taxa have been commercially used as effective biofertilizers [20]. The benefits of microbial inoculants over chemical inoculants are numerous [21,22], i.e., they are safe, sustainable sources of renewable nutrients that are essential for soil health and life [23,24]. They can be good at N_2_-fixation, phosphate solubilization, and plant growth stimulation, or they can have a mix of these abilities [25,26]. Diazotrophic bacterial endophytes should be able to colonize host plant roots adequately, generate a suitable rhizosphere for plant growth improvement, and increase the bioavailability of N, P, K, and antagonistic activity as bio-fertilizers [27,28].

Nitrogen availability normally restricts ecosystem production, and N_2_-fixation, which is exclusive to prokaryotes, is a significant main source of input that supports food webs. Diazotrophic microbes are found in the great bulk of Earth’s ecosystems. Diazotroph diversity and community composition are directly associated with ecosystem function. Moreover, environmental variability and previous methodological limitations have frequently limited our capacity to relate individual diazotroph groups to ecosystem performance. Because the great majority of diazotrophs have yet to be cultivated, cultivation-independent molecular methods are critical for comprehending community dynamics, and the *nifH* gene is the most commonly utilized molecular surrogate for N_2_-fixation capacity [29]. The identification of *nif* gene sequences encoding Nase is the primary foundation for characterizing these bacteria. The *nifH* gene, which codes for the iron protein of the nitrogenase enzyme, is the most widely used genetic marker for studying uncultured N-fixing microbes in nature. In this study, we used polF/polR primers because the selection of *nifH* primers might affect the recovery of *nifH* sequences owing to variance in coverage, and this primer was previously utilized in a study of the diazotrophic microbial community in sugarcane [30]. For example, whereas Zf/Zr primers show stronger theoretical recovery of *nifH* diversity, polF/polR primers function better on the bench [31].

The microbial endophyte is a promising area of study in plant biology and a substantial amount of investigation about endophytic microbes and their positive effect on plants has been established [32,33]. The increased scientific literature over the last three decades demonstrates that researchers are more concerned with the study of endophytic microorganisms [34,35,36]. Endophytic bacterial inoculants as microbial biofertilizer alternatives would be a potential source of chemical fertilizers and have industrial applications. Plant–endophytic microbe interactions can be broadly characterized as neutral, useful (positive), or damaging (negative), while various kinds of connections are known to occur in ecological relationships [37]. The utilization of root bacterial endophytes in the development of bioinoculants has been successful, and their use in current agricultural methods appears promising [38]. Numerous studies have reported endophytic bacteria that can promote the growth of plants such as wheat, rice, canola, potato, tomato, sugarcane, and many more [14,39,40,41]. However, endophytes’ diverse host range makes them an effective tool in agricultural biotechnology. As a result, endophytes have huge potential for application as biofertilizers and biopesticides in the growth of a sustainable, safe, and efficient agricultural systems.

Metagenomics is a revolution against the drawbacks of culture-based methods, which have seen significant growth in use in recent decades. Unfortunately, owing to the boundaries of the culture-based technique, there are still many microorganisms that cannot be identified and cultured. Therefore, in this method without a culturing technique in the laboratory, DNA is extracted directly from environmental materials, and this technique can help researchers overcome these issues [42]. This article also discusses the potential of the metagenomics method to facilitate the growth of uncovering novel *nifH* gene producing microbes from sugarcane yet uncultured, which helped researchers to isolate and identify those that are useful for agriculture.

To investigate the culturable diversity of endophytic diazotrophic bacteria in root, stem, and leaf tissues of different sugarcane species, a total of 345 endophytic bacteria were obtained in different mediums from different tissues of sugarcane. Diazotrophic microbial biofertilizers can fix atmospheric N_2_ through the BNF process and solubilize nutrients required by the plants. Approximately 50% of land N_2_ is derived from BNF (agricultural and natural), which occurs primarily in *Rhizobium* legume symbiosis and produces inorganic N_2_ primarily in the form of NH_4_^+^ [43]. Because of their capacity to convert N to NH_4_^+^, a form of N_2_ readily utilizable by the host plant, there is growing interest in determining the physiological and molecular mechanisms underlying this plant–microbe interaction [44]. Because of its low cost, simplicity, and high throughput potential, the ARA (C_2_H_2_-C_2_H_4_), that measures the activity of the N_2_-fixing nitrogenase enzyme in reducing acetylene to ethylene [45], is a common method for quantifying BNF in both symbiotic and free-living BNF niches for culturable microbiomes. This technique rests on fact that the N_2_-fixing enzyme (nitrogenase) system can also reduce C_2_H_2_-C_2_H_4_, through the same pathway as that of N_2_-fixation. The ARA is based on Nase’s ability to convert acetylene to ethylene, a molecule that can be easily quantified using gas chromatography [46,47]. Our cultivable microbes such as *Bacillus velezensis*, *Burkholderia* sp., *Enterobacter roggenkampii*, *Herbaspirillum aquaticum*, *Kosakoniaoryzae*, *Lelliottia nimipressuralis*, *Microbacterium* sp., *Pantoea dispersa*, and *Pseudomonas putida* showed maximum N_2_-fixation in Table 1. Nitrogenase-producing bacteria have also been isolated from sugarcane, according to previous investigations [11,13,14,48,49,50,51].

To uncover the deeper community structure and abundance of N_2_ fixers in sugarcane, we used high-throughput sequencing methods rather than traditional cultivation approaches and clone libraries. Endophytic bacterial diazotrophic communities of distinct sugarcane species in different tissues are phylogenetically characterized using experimentally obtained *nifH* similarity cutoffs at the species, genus, and family levels. The high quantities of sequence information produced by high-throughput technology open up new research opportunities and also pose new data-processing challenges. *nifH* gene sequence data revealed 319766 OTUs assigned to different diazotrophic bacterial species colonizing the different sugarcane *Saccharum* species in different tissues studied. Analysis of these OTUs showed that *S. sinense* Roxb. cv Uba confined the largest number of diazotrophic communities compared to *S. barberi* Jesw. cv Pansahi > *S. robustum* > *S. spontaneum* > *S. officinarum* L. cv. Badila species. As a result, the microbial community composition and abundance within sugarcane plants, as well as the number of bacterial endophytes, can differ depending on soil type, management of fertilizer application, season, plant type, plant growth stage, and variety [52,53,54]. 

Our study reported the genus level of the most abundant N_2_-fixing microbes are Saccharum, *Ideonella*, *Kosakonia*, *Bradyrhizobium*, *Burkholderia*, *Azospirillum*, *Herbaspirillum*, *Rhizobium*, *Sorghum*, *Pseudacidovorax*, *Enterobacter*, *Stenotrophomonas*, *Methylobacterium*, *Desulfocapsa*, *Klebsiella*, *Dechloromonas*, *Ruficoccus*, *Rhodoplanes*, *Ampullimonas*, *Desulfovibrio*, *Paenibacillus*, *Geobacter*, *Vitreoscilla*, *Scytonema*, *Sulfuricurvum*, *Magnetospirillum*, *Sphingomonas*, *Rhodopseudomonas*, *Xanthobacter*, *Methyloversatilis*, *Panicum*, *Corallococcus*, *Lachnoclostridium*, *Azoarcus*, *Azohydromonas*, *Thermincola*, *Paraburkholderia*, *Zea*, *Tolumonas*, *Rhodococcus*, *Pleomorphomonas*, *Rubrivivax*, *Beijerinckia*, *Aegilops*, *Ensifer*, *Spinacia*, *Phleum*, *Calenema*, *Azonexus*, *Hymenobacter*, *Azorhizobium*, *Pseudodesulfovibrio*, *Sinorhizobium*, *Escherichia*, *Bacillus*, and *Exophiala*. Among these some are well-known examples of N_2_-fixing bacteria; Azotobacter, Azospirillum, Herbaspirillum, Gluconacetobacter, Burkholderia, and Paenibacillus are well-known bacteria with a long history of use in sugarcane and cereal crops for growth and yield [12,55,56,57,58,59]. For many of these genera, little is known about their biological role, particularly in terms of N_2_-fixation. However, there are regular attempts to recover them in sugarcane for culture-dependent approaches and studies based on ARA and nifH gene sequencing.

In comparison to the traditional clone library method, the next-generation sequencing technology was effectively applied to produce trustworthy results in the analysis of diazotroph community composition and diversity [60]. The bacterial community structure of the root, stem, and leaf portions was substantially altered by all sugarcane genotypes based on the α and β diversities. Through alpha diversity examination of five different sugarcane species (*p* < 0.03), we found that the bacterial population diversity and abundance were considerably higher in *S. barber**i* Jesw. cv Pansahi > *S. robustum* > *S. spontaneum* > *S. officinarum* L. cv. Badila (Table 3), which meant that the association between bacteria and the sugarcane tissues was more complicated and the ecosystem was more stable. The outcomes of beta-diversity analysis revealed that the diazotrophic bacterial communities of each component were significantly different, especially in the root of all sugarcane species than leaf and stem components, and these two compartments indicated more directly related patterns and belonging to the same phyla (Figure 5). Each treatment generated a distinct community structure, according to hierarchical cluster examination of similarity and PCoA examination of the bacterial communities. In all treatments, the dominating and responsive taxa were Proteobacterium, Streptophyta, Actinobacteria, Cyanobacteria, Firmicutes, and Verrucomicrobia, comparable to earlier research in sugarcane soils [61]. The relative abundances of Proteobacteria were much greater in root tissues than in stem and leaf tissues, according to our findings. Additionally, the dominance of α and β-Proteobacteria is common, as they are widespread in all sugarcane species. To discover microbial population distribution changes among all tissues of distinct sugarcane species, researchers used relative abundance and variation at the phylum level, as well as LEfSe at each taxonomic level from phylum to genus [62]. 

This is the first study to characterize the EDBCs composition in the root, stem, and leaf tissues of five *Saccharum* species grown in the sugarcane field using high-resolution community profiling of original species and commercial cultivars. We observed that *Saccharum*, *Bradyrhizobium*, *Burkholderia*, *Azospirillum*, *Rhizobium*, *Sorghum*, *Pseudacidovorax*, *Stenotrophomonas*, *Methylobacterium*, *Klebsiella*, *Dechloromonas*, *Ampullimonas*, *Desulfovibrio*, *Methyloversatilis*, *Panicum*, *Lachnoclostridium*, *Azohydromonas*, *Pleomorphomonas*, *Rubrivivax*, and *Beijerinckia* are the top 20 genera with the highest abundance in all sugarcane species. Out of these, some genera are well known for N_2_-fixation, i.e., *Bradyrhizobium*, *Burkholderia*, *Azospirillum*, *Rhizobium*, *Stenotrophomonas*, and *Klebsiella* in sugarcane. Therefore, further, we must discover and identify new genera to fix N_2_ in sugarcane and other crops to produce bio-based agricultural products that will increase crop production.

## 4. Material and Methods

### 4.1. Analysis of Endophytic Diazotrophic Bacterial Diversity through Culture-Dependent Techniques

#### 4.1.1. Study Site and Sample Collection

Healthy sugarcane plant samples were selected for this study, and 25 plants were collected from five different sugarcane species, i.e., *Saccharum officinarum* L. cv. Badila, *S. barberi* Jesw. cv Pansahi, *S. robustum*, *S. spontaneum*, and *S. sinense* Roxb. cv Uba. Samples from each sugarcane species were collected from five random sampling points and shifted in the laboratory. The sampling area has a humid subtropical climate; with an average annual rainfall of 1000–2800 mm. All samples were carefully uprooted and gently washed with tap water, and again washed with autoclaved distilled water to remove all soil and dust particles as well as dead tissues on the plants. The plant samples were sliced into small pieces and composed separately from the root, leaf, and stem; then placed in liquid nitrogen and stored at 20 °C for diversity analysis of *nifH* microbial communities. All of these sugarcane plant samples were obtained from Guangxi Academy of Agricultural Sciences, Sugarcane Research Centre, located at (latitude 22°49′1.21″ N, longitude 108°21′59.55″ E), Nanning, Guangxi, China. A total of 15 samples from different varieties of sugarcane plant tissues such as root, stem, and leaf were selected for the analysis of microbial diversity of *nifH* gene-producing micro-organisms.

#### 4.1.2. Isolation of Endophytic Diazotrophs from Sugarcane Plants

For the isolation of N-fixing endophytic bacteria, 10 different mediums were used, namely, Ashbys Glucose Agar, Ashbys Mannitol Agar, Burks Medium, Jensens Agar, Nutrient Agar, Luria-Bertani Agar, Yeast Mannitol Agar, DF salts minimal medium, Pikovskayas Agar, and Chrome azurol-S agar. After sterilization, 1 mL of sterile 5% sucrose solution was combined with one gram of fresh root pieces and crushed. The isolation of the endophytic diazotroph bacterial strains process was completed by Guo et al. [51], and morphologically distinct strains developed from the root were picked and purified from the bacterial colonies. All endophytic bacterial strains were stored at −20 °C in a 20% glycerol solution.

#### 4.1.3. Estimation for Nitrogenase Activity by Acetylene Reduction Assay

The potential of each endophytic diazotroph isolate to fix N_2_ was assessed using acetylene reduction assay (ARA), which was previously illustrated by Hardy et al. [45], and the process was completed by Singh et al. [63].

#### 4.1.4. DNA Extraction, 16S and *nifH* Gene Amplification of Endophytic Microbes

The genomic DNA of all endophytic strains was extracted with a DNA isolation kit (CWBIO, Beijing- China). All isolate genomic DNA templates were utilized to amplify the 16S rRNA gene using universal primers [64], and a conserved area of the *nifH* gene fragment according to Poly et al. [65] by using the PCR method. A purified PCR product was sequenced commercially by Sangon Biotech (Shanghai, China).

### 4.2. Analysis of Endophytic Diazotrophic Bacterial Communities by Culture-Independent Technique 

#### 4.2.1. DNA Extraction

All collected samples from different tissues such as root, stem, and leaf of five different sugarcane species were processed separately. The total genomic DNA of all samples was extracted directly with QIAamp DNA Stool Mini Kit (QIAGEN, Hilden, Germany), according to the manufacturer’s directions. The quality of DNA was checked with agarose (1.2%) gel electrophoresis (Appendix A). The DNA concentration for all samples was quantified by FTC-3000™ and Real-Time PCR cycler and stored at −20 °C for further use.

#### 4.2.2. Construction of Sequencing Libraries and PCR Amplification

The structure and diversity of endophytic diazotrophic microbial communities (EDMCs) were analyzed through Illumina Miseq 2 × 300 BP high throughput sequencing method (Micro-based Biotechnology Co., Ltd., Shanghai, China). A two-step PCR amplification approach was used to create the library. First, a specific primer (inner primer) was used to amplify the target fragment. Following that, the target fragment was gel recovered, and the recovered product was employed as a template for secondary PCR amplification (outer primer). The purpose is to add the Illumina sequencing platform’s adaptor or linker, sequencing primer, and barcode to both ends of the target fragment. 

The *nifH* gene of EDBCs was amplified using the universal N-fixing bacterial primers Pol-F (5′-TGCGAYCCSAARGCBGACTC-3′) and Pol-R (5′-ATSGCCATCATYTCRCCGGA-3′) as earlier described by Poly et al. [65]. PCR for primer specificity testing was completed in 50 µL of the final volume containing 10 µL; 5× PCR buffer, 1 µL; dNTP mix (10 mM), 1 µL; both primers (forward and reverse) (10 µM), 1U; Taq DNA polymerase (Phusion Ultra Fidelity DNA Polymerase), 2 µL; extracted DNA template (5–50 ng), and total volume of 50 µL with ddH2O. PCR amplifications were accomplished with an initial denaturation at 94 °C for 2 min that was followed by 35 cycles of denaturation at 94 °C for 30 s, 55 °C annealing for 35 s, and extension at 72 °C for 5 min, and a 5 min extension at 72 °C following the last cycle. PCR product (4 µL) was assayed by gel electrophoresis on a 1.2% agarose gel (Appendix A).

The PCR products with 2–3 bands were detected, and then we purified the band to become a single fragment length of the expected size of 360 bp. The PCR amplification product was recovered by agarose (2%) gel electrophoresis. All samples of *nifH* gene PCR products were recovered using AxyPrep DNA Gel Recovery Kit (AXYGEN’s). The recovered product was then subjected to the second round of PCR amplification. PCR for primer specificity testing was carried out in a total volume of 50 µL containing 8 µL; 5× PCR buffer, 1 µL; dNTP mix (10 mM), 1.0 µL; both primers (forward and reverse) (10 µM), 0.8 U; Taq DNA polymerase (Phusion Ultra Fidelity DNA Polymerase), 5 µL; template, and 33.2 µL ddH_2_O. For PCR amplifications, initial denaturation at 94°C for 2 min was followed by 8 cycles of denaturation at 94 °C for 30 s, 56 °C annealing for 30 s, and extension at 72 °C for 30 s. After the previous cycle, a 5 min final extension at 72 °C was performed. PCR product (4 µL) was assayed by gel electrophoresis on a 1.2% agarose gel. After the second PCR amplification, electrophoresis results showed the bands are single and the brightness is moderate. The PCR amplification product was recovered by agarose (2%) gel electrophoresis. Appendix A shows the electrophoresis results of *nifH* gene amplification after the final PCR, and the bands are single and bright. It was recovered using AXYGEN’s AxyPrep DNA Gel Recovery Kit and the recovered product was subjected to quantification by FTC-3000^TM^ RT-PCR. The amplicons were sequenced with Illumina MiSeq 2 × 300 bp (Illumina Inc., San Diego, CA, USA) after the samples were prepared by combining them in equimolar ratios.

#### 4.2.3. Diazotrophic Microbial Community Profile, Data Processing, and Bioinformatic Analysis

The paired-end (PE) reads obtained from the MiSeq machine were separated using barcode information, and the reads were checked for quality using FastQC [66]. The PE reads were merged into one contig and quality filtered; the lengths of reads with an average quality < 20 were filtered off using Trimmomatic [67] keeping the default parameter. The subsequent sequences were firstly treated to remove the low-quality data and recover the syncretic rates of the remaining sequences, and the maximum mismatch percentage acceptable by the overlay region of the splicing sequence was 0.2.PE and sequences were assembled using FLASH software [68]. Sequences containing short segments, low complexity, and low-quality nucleotides were removed by PRINSEQ-lite 0.19.5 [69]; errors were modified with the Pre. cluster tool [70]. Further, the spliced sequence was subjected to quality control and filtering using MothurV.1.39.5 [71]. Finally, the optimized sequence was used for subsequent clustering OTU and species information analysis.

USEARCH for operational taxonomic units (OTU) cluster analysis was performed on the high-quality tags sequence obtained. OTUs were clustered with ≥97% similarity using UPARSE, and the taxonomic based classification was achieved on ribosomal DNA sequences with Ribosomal Database (RDPII) with a classification threshold of 0.8 [72,73]. The relative abundance of OTU was compared across all the sample sequences obtained by USEARCH. This representative sequence was then searched to classify Bacterial and Archaeal species using databases of green genes, SILVA 128, SILVA 119; UNITE [74]. For sequence annotation, Mothur was employed with a confidence threshold of 0.8. The OTU representative sequences taxonomically identified were compared with the NCBI GenBank database for the BLASTn suit search program. To facilitate the analysis of the percentage corresponding to each OTU in each sample, the sample OTU distribution and the species information were homogenized, and all were homogenized to 100%. The information in the OTU comprehensive classification table was extracted according to the six levels of Phylum, Class, Order, Family, Genus, and Species, and the relative abundance percentage of each sample at different classification levels was counted separately.

The EDMCs analysis was performed using the rank abundance method by counting the number of sequences contained in each OTU through a single sample diversity analysis (Alpha diversity), including a series of statistical analysis indices to estimate species abundance and diversity of ecological communities. The alpha diversity of the EDMCs was measured using the following indices: Shannon’s diversity index (species diversity) [75,76], Simpson index (community diversity), Chao index (species richness) [77], and ACE index, the abundance-based coverage estimator (species richness) [78].

Relationships between the beta diversity estimations of EDMCs were calculated with the weighted UniFrac analysis [79] by sample distance heat map intensity [80]. Beta diversity heat map using R (3.4.1) software heatmap was included in the plot. The distance matrix obtained by UniFrac analysis can be used in a variety of analytical methods and can be used to visualize the similarities and differences in microbial evolution in different environmental samples by multivariate statistical methods. PCoA (principal coordinates analysis), UPGMA (unweighted pair group method with arithmetic mean), and non-metric multidimensional scale (NMDS) were used to evaluate the differences amongst the microbial communities, measured significant at *p*-values less than 0.001. Representational difference analysis (RDA) is based on a linear model that was created to determine the relationship between environmental factors, samples, and flora, or the relationship between the diversity of EDMCs. Characterization of EDMCs features in the individual group was completed by using the linear discriminant analysis (LDA) and effect size (LEfSe) analyses were performed using the LEfSe tool for biomarker discovery, which emphasizes the statistical significance and biological relevance [81,82]. The relative abundances of endophytic diazotrophs were expressed as percentages, and differences in EDMCs abundance were analyzed by LDA Effect Size (LEfSe). Through the normalized relative abundance matrix, the LEfSe analysis uses the Kruskal–Wallis (KW) to detect sum-rank test features with significantly different abundances between the assigned taxa and performs the LDA to estimate the effect size of each feature [83]. The values are presented as a means of least-square with standard errors. Additionally, differences were considered significant at *p*-values less than 0.05. Similarity analysis of ANOSIM was used to test differences between groups (two or more groups) in microbial community composition. Core microbiome analysis was performed on shared OTU of the samples and species represented by the OTU using R (3.4.1).

## 5. Conclusions

Many diazotrophic microbial species colonize different tissue of plants, and isolation of such useful plant MCs presents significant difficulties in the laboratory and in vivo experimental investigation. Therefore, the present work described the variation in EDBCs in different tissues (root, stem, and leaf) of five *Saccharum* species. This new approach provides evidence of diazotroph ecology with MiSeq high-throughput sequencing by targeting the *nifH* gene, and bioinformatics analysis. Results showed a significant difference in diazotrophic communities of all sugarcane species and sugarcane roots had higher species richness as compared to leaf and stem tissues, confirming that root tissues of sugarcane are a good source of novel endophytic diazotrophic bacteria. At the phylum level, *Proteobacteria* was the most occupied phyla in root compared to leaf and stem tissues. Additionally, these results present a well-structured *nifH* microbial community in all plant components, which is the first report of this type for a functional community, specifically EDBCs in five *Saccharum* species. Considering the present finding, future research could focus on the identification and application of these useful plant growth-promoting endophytic bacteria, particularly diazotrophs, in *Saccharum* species. It is vital to fill the information gap on microbial populations and sugarcane interactions, and potent N_2_-fixing EDBCs may be applicable for sustainable sugarcane cultivation.

## Figures and Tables

**Figure 1 ijms-23-06242-f001:**
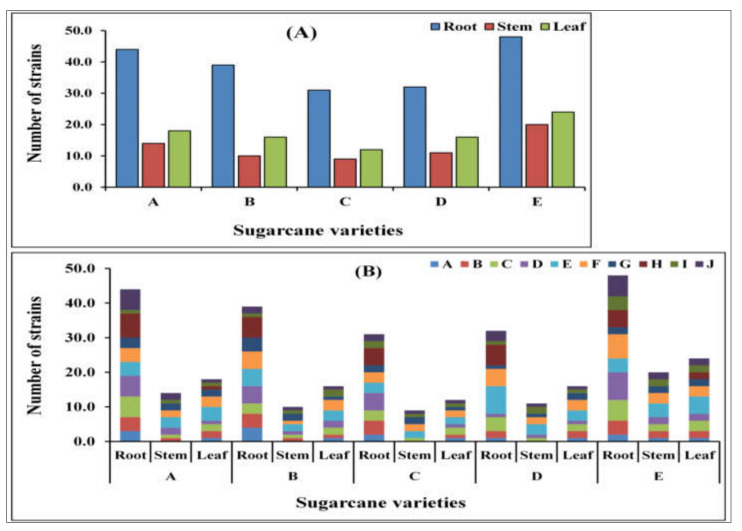
The total number of isolated endophytic bacterial strains. (**A**) Total number of bacterial strains isolated from different sugarcane species in different tissues, i.e., root, stem, and leaf. A. *Saccharum officinarum* L. cv. Badila, B. *S. barberi* Jesw.cv Pansahi, C. *S. robustum*, D. *S. spontaneum*, and (E) *S. sinense* Roxb.cv Uba). (**B**) Number of individual isolates from different tissues (root, stem, and leaf) in the different media: A. Ashby’s Glucose Agar, B. Ashby’s Mannitol Agar, C. Burk’s Medium, D. Jensen’s Agar, E. Nutrient Agar, F. Luria–Bertani Agar, G. Yeast Mannitol Agar, H. DF salts minimal medium, I. Pikovskaya’s Agar, and J. Siderophore medium of sugarcane species.

**Figure 2 ijms-23-06242-f002:**
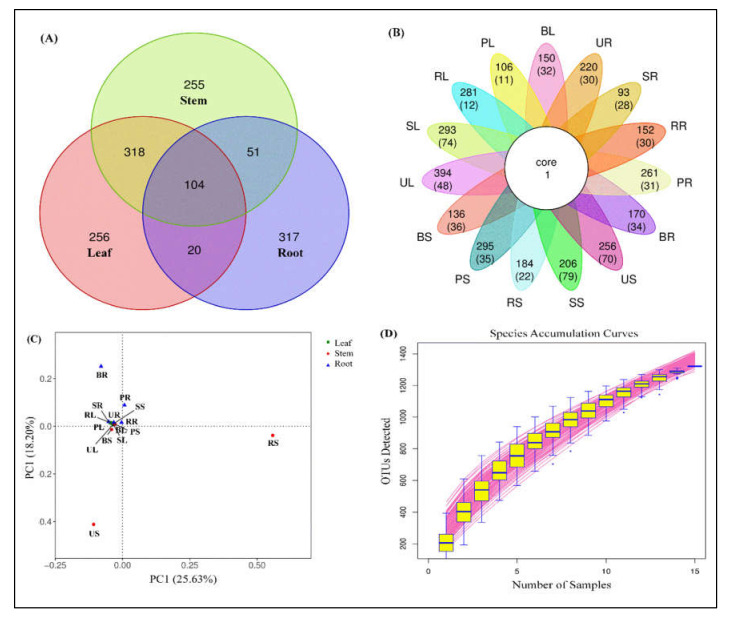
Operational taxonomic unit (OTU) clustering among five sugarcane species. (**A**) Venn diagram showing several unique and shared OTUs in combined plant samples. (**B**) Petal map displaying the number of unique and shared OTUs, the outermost circle represents the sample name; the petals include two rows of numbers, with the number of all OTUs contained in each sample and the number of OTUs unique to each sample in the following brackets; the white circle in the center represents the core OTU quantity. (**C**) Principal component analysis (PCA) showing the percentage of each OTU in each sample, and (**D**) species accumulation curve describing the increase in species as the sampling amount increases and an effective tool for understanding the species composition of surveyed plots and predicting species richness.

**Figure 3 ijms-23-06242-f003:**
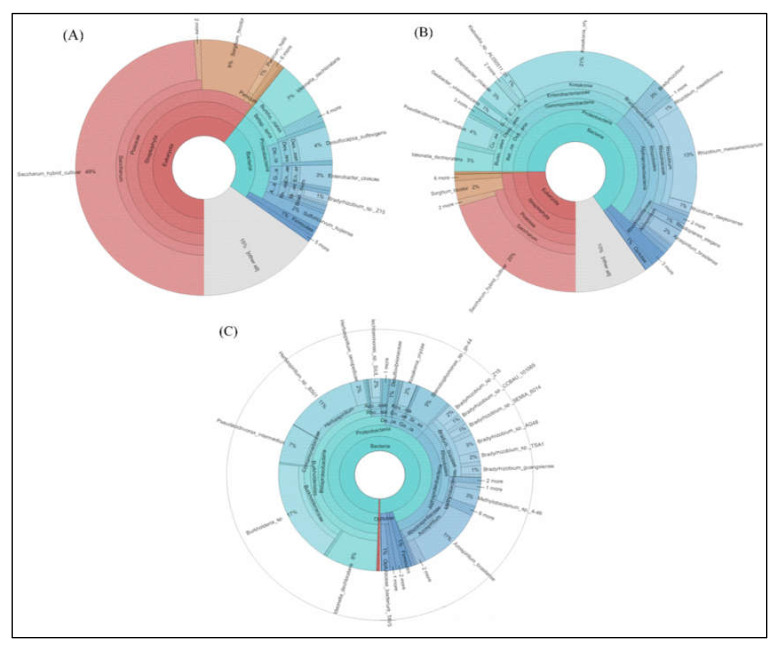
Krona software results show results of species annotations of (**A**) leaf, (**B**) stem, and (**C**) root samples. In the display results, the circles represent different classification levels from the inside to the outside, and the size of the fan represents the relative proportion of different OTU annotation results.

**Figure 4 ijms-23-06242-f004:**
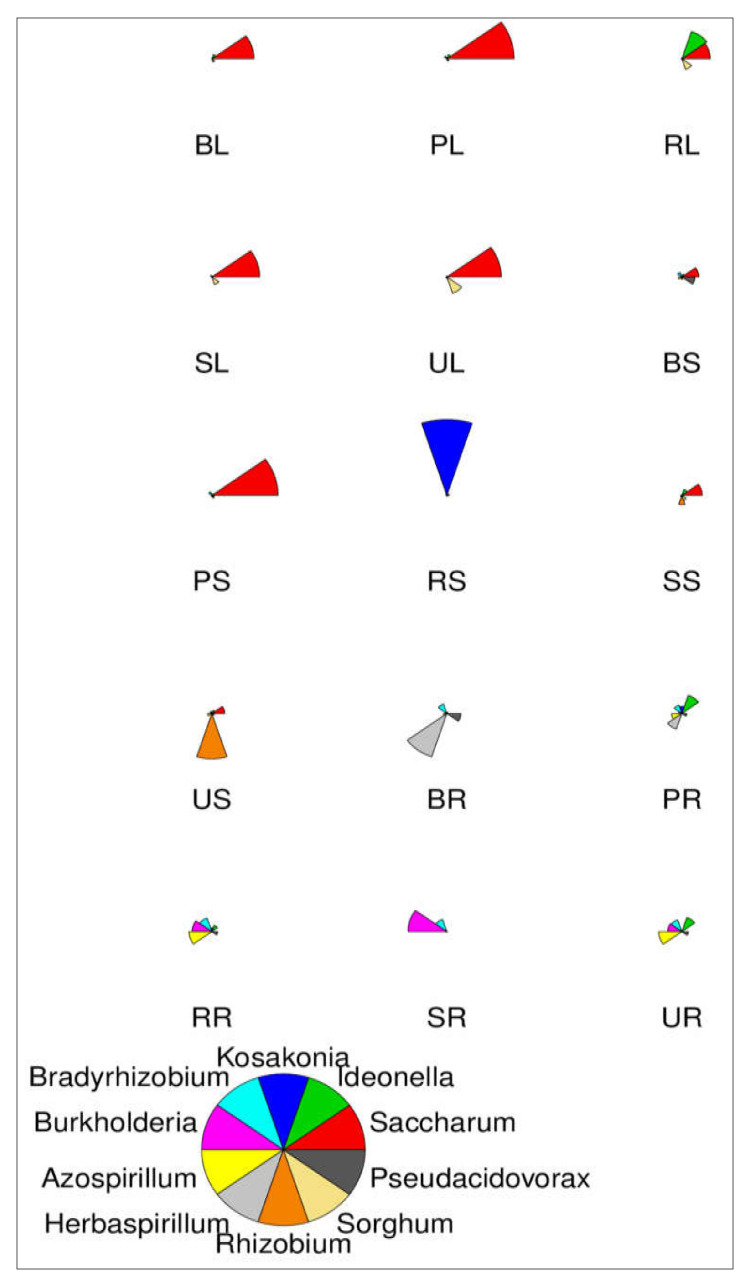
Star map showing species abundance of the top 10 most abundant genera. The fan shape in each star chart represents a species, which is distinguished by different colors. The radius of the fan is used to represent the relative abundance of the species. The longer the radius of the fan, the higher the abundance, and the lower the abundance.

**Figure 5 ijms-23-06242-f005:**
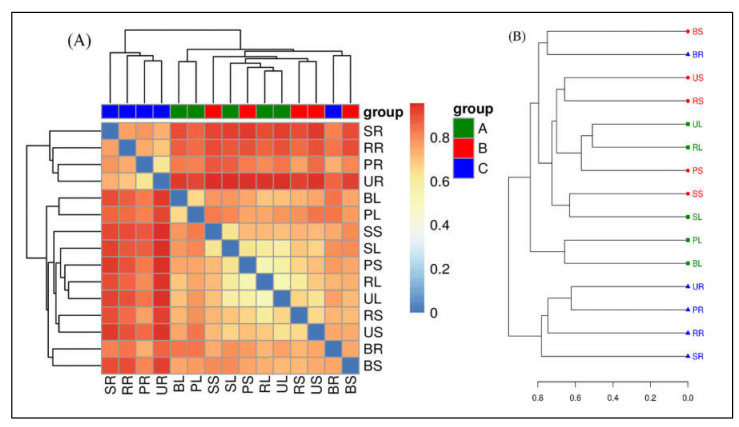
Beta-diversity analysis. (**A**) Matrix heat map of UniFrac; Beta-diversity matrix heatmap visualizes the Beta-diversity data and graphically clusters the samples and samples with similar beta diversity are clustered together to reflect similarities between samples. (**B**) UniFrac multi-sample similarity tree assessment; the distance matrix derived from Unifrac analysis is used in a wide range of analysis methods. The non-weighted group averaging method unweighted pair group method with arithmetic mean (UPGMA) in hierarchical clustering is used to construct graphical visualization processing such as a phylogenetic tree, that can visually show the similarity and differences in microbial evolution in different environmental samples.

**Figure 6 ijms-23-06242-f006:**
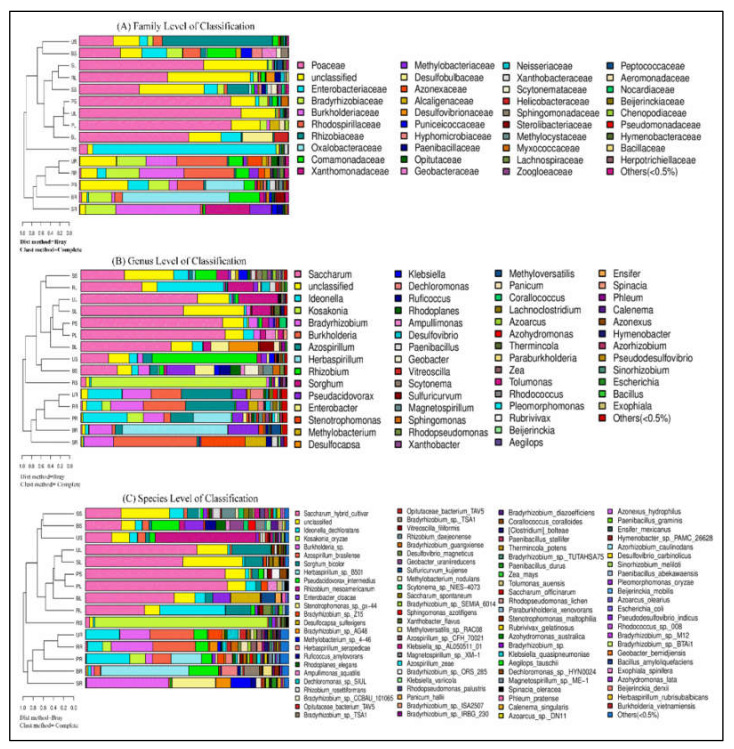
Combined analysis of similarity tree and histogram. (**A**) Family level of classification, (**B**) genus-level of classification, and (**C**) species-level of classification. On the left is the hierarchical clustering analysis (Bray–Curtis algorithm) based on community composition among the samples and on the right is the histogram of the community structure of the samples.

**Figure 7 ijms-23-06242-f007:**
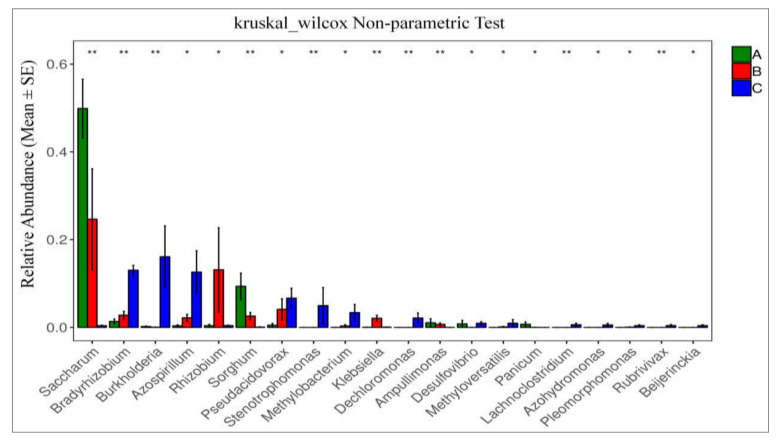
Species difference analysis at each classification level. Each column is the relative abundance of the different species. The error line is standard error, where * represents 0.01 < *p* < 0.05 and ** represents 0.001 < *p* < 0.01.

**Figure 8 ijms-23-06242-f008:**
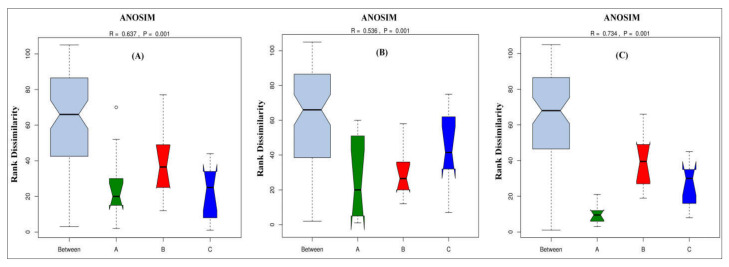
Anosim analysis of (**A**) leaf, (**B**) stem, and (**C**) root. Anosim analysis is used to determine whether a group is meaningful by determining whether differences between groups are significantly greater than intra-group differences. If the R-value is greater than 0, the difference between the groups is greater than the intra-group difference, and the grouping is more reasonable. If the R-value is less than 0, the difference between the groups is smaller than the intra-group difference, and the group is defective. The greater the *p*-value, the greater the difference between groups.

**Figure 9 ijms-23-06242-f009:**
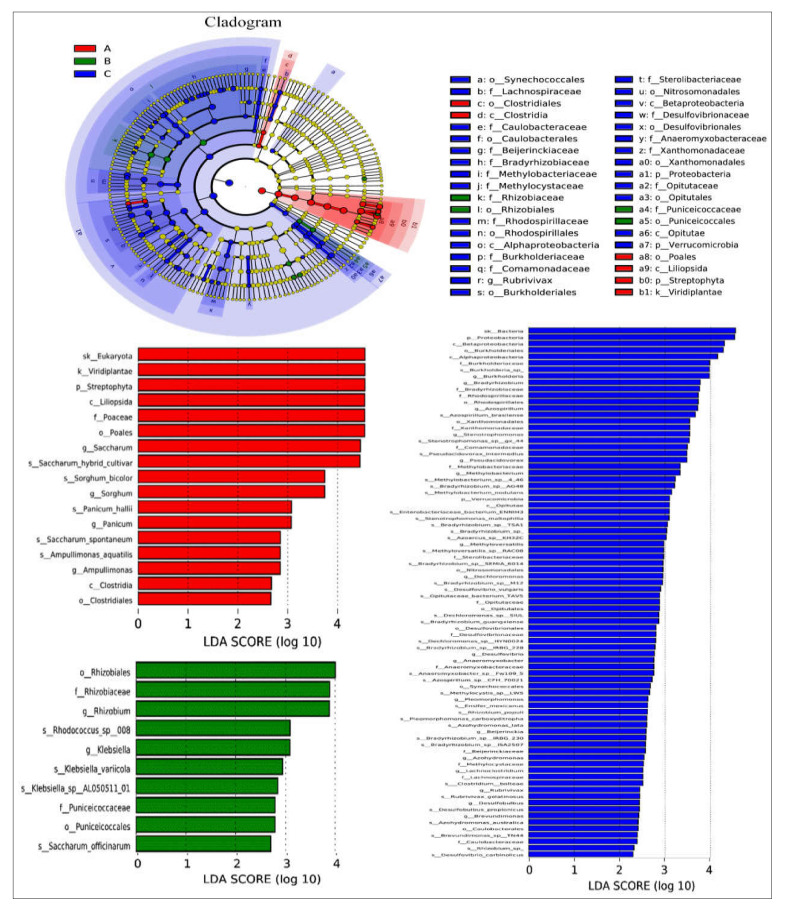
Analysis of community differences between LEfSe groups. LEfSe uses linear discriminant analysis (LDA) to estimate the effect of the abundance of each component (species) on the different effects.

**Table 1 ijms-23-06242-t001:** PCR amplification of the *nifH* gene and nitrogenase activity of selected isolates through acetylene reduction assay.

Strain Name	*nifH* Gene Amplification	Accession Number	ARA (nmoL C_2_H_4_ mg Protein h^−1^)
*Achromobacter* sp.	+	-	12.415 ± 0.225 ^r^
*Achromobacter xylosoxidans*	+	-	17.133 ± 0.310 ^l^
*Atlantibacter hermannii*	+	-	13.258 ± 0.240 ^q^
*Bacillus* sp.	+	-	9.274 ± 0.168 ^v^
*Bacillus amyloliquefaciens*	+	-	11.161 ± 0.202 ^t^
*Bacillus velezensis*	+	-	21.800 ± 0.395 ^g^
*Burkholderia* sp.	+	-	26.457 ± 0.479 ^b^
*Burkholderia cenocepacia*	+	-	15.778 ± 0.286 ^mn^
*Burkholderia gladioli*	+	-	18.498 ± 0.335 ^jk^
*Chryseobacterium* sp.	+	-	14.513 ± 0.263 ^o^
*Enterobacter* sp.	+	MT649072	11.173 ± 0.191 ^u^
*Enterobacter asburiae*	+	MT649071	08.234 ± 0.222 ^rs^
*Enterobacter cloacae*	+	MT649070	14.724 ± 0.267 ^o^
*Enterobacter roggenkampii*	+	MT649078	29.709 ± 0.538 ^a^
*Enterobacter tabaci*	+	-	13.309 ± 0.241 ^q^
*Herbaspirillum aquaticum*	+	-	24.781 ± 0.449 ^cd^
*Kosakonia oryzae*	+	MZ497007	10.285 ± 0.422 ^f^
*Lelliottia nimipressuralis*	+	-	24.158 ± 0.438 ^e^
*Lysinibacillus macroides*	+	-	13.891 ± 0.252 ^p^
*Metakosakonia* sp.	+	-	6.163 ± 0.112 ^w^
*Microbacterium* sp.	+	-	24.570 ± 0.445 ^de^
*Pantoea* sp.	+	-	15.356 ± 0.278 ^n^
*Pantoea agglomerans*	+	MZ502262	10.082 ± 0.183 ^u^
*Pantoea ananatis*	+	MZ502260	18.645 ± 0.339 ^j^
*Pantoea dispersa*	+	MZ502257	15.305 ± 0.422 ^f^
*Pseudomonas* sp.	+	-	17.243 ± 0.312 ^l^
*Pseudomonas aeruginosa*	+	MW027642	12.626 ± 0.229 ^r^
*Pseudomonas chlororaphis*	+	-	18.156 ± 0.329 ^k^
*Pseudomonas plecoglossicida*	+	-	18.919 ± 0.343 ^ij^
*Pseudomonas putida*	+	-	25.202 ± 0.456 ^c^
*Pseudomonas koreensis*	+	-	15.567 ± 0.282 ^n^
*Pseudomonas taiwanensis*	+	-	12.215 ± 0.221 ^rs^
*Rhizobium* sp.	+	-	13.258 ± 0.240 ^q^
*Serratia* sp.	+	-	18.919 ± 0.343 ^ij^
*Serratia marcescens*	+	-	14.513 ± 0.263 ^o^
*Sphingomonas azotifigens*	+	-	17.614 ± 0.319 ^l^
*Sphingomonas echinoides*	+	-	19.401 ± 0.351 ^hi^
*Sphingomonas trueperi*	+	-	16.189 ± 0.293 ^m^
*Staphylococcus arlettae*	+	-	11.793 ± 0.214 ^s^
*Stenotrophomonas* sp.	+	-	13.258 ± 0.240 ^q^
*Stenotrophomonas maltophilia*	+	-	12.415 ± 0.225 ^r^
*Xanthomonas sacchari*	+	-	19.541 ± 0.354 ^h^

Different lowercase letters present a significant difference *p* ≤ 0.05.

**Table 2 ijms-23-06242-t002:** Optimized sequence data used for subsequent clustering operational taxonomic units and species information analysis.

Sugarcane Variety	Plant Parts	Number of Valid Sequences	Optimized Number of Sequences	Optimized Number of Sequence Bases	Optimized Sequence GC Content (%)	Optimized Average Sequence Length	Optimized Sequence Length Range
*S. officinarum* L. cv. Badila	BL	32,624	25,686	7,891,580	50.374	307	200→541
	BS	18,573	15,532	5,212,342	58.127	336	200→502
	BR	36,189	27,995	10,001,455	61.731	357	201→469
*S. barberi* Jesw. cv Pansahi	PL	19,097	12,937	3,714,556	42.949	297	200→451
	PS	21,840	15,093	4,309,289	44.523	296	201→541
	PR	37,161	21,601	7,752,193	61.927	359	202→405
*S. robustum*	RL	27,071	11,358	3,669,350	56.315	323	200→470
	RS	37,944	24,191	8,616,963	55.621	356	201→470
	RR	37,700	30,476	10,925,709	62.800	359	202→393
*S. spontaneum*	SL	35,409	21,038	5,897,235	48.693	290	200→471
	SS	37,892	19,058	5,851,645	56.441	307	202→476
	SR	36,135	31,283	11,253,996	62.336	360	261→470
*S. sinense* Roxb. cv Uba	UL	31,562	16,004	4,724,294	48.035	295	200→478
	US	36,943	25,611	8,610,553	57.739	336	200→478
	UR	36,005	21,903	7,870,912	63.059	359	226→369

**Table 3 ijms-23-06242-t003:** Diversity indices for *nifH* gene analysis of endophytic diazotrophic bacterial communities in different tissues of five sugarcane species.

Sugarcane Variety	Sample Code	Reads	OTU	Ace	Chao	Coverage	Shannon	Simpson	Sobs	PD Whole Tree
*S. officinarum* L. cv. Badila	BL	25,686	150	231.49	193.50	0.998832	3.78	0.0445	150.00	53.66
	BR	27,995	170	186.34	198.11	0.999178	2.52	0.2148	170.00	45.02
	BS	15,532	136	160.16	155.12	0.998326	3.49	0.0503	136.00	55.06
*S. barberi* Jesw. cv Pansahi	PL	12,937	106	125.54	121.11	0.998686	3.33	0.0808	106.00	33.68
	PR	21,601	261	279.13	286.37	0.998657	4.08	0.0426	261.00	26.27
	PS	15,093	295	446.16	395.04	0.995097	4.52	0.0172	295.00	79.38
*S. robustum*	RL	11,358	281	344.88	346.21	0.994541	3.79	0.0998	281.00	73.35
	RR	30,476	152	160.49	174.00	0.999606	3.71	0.0587	152.00	19.59
	RS	24,191	184	222.06	234.17	0.998222	1.62	0.363	184.00	57.39
*S. spontaneum*	SL	21,038	293	411.03	371.00	0.997528	4.34	0.0354	293.00	88.57
	SR	31,283	93	96.90	95.50	0.999840	2.53	0.1437	93.00	11.37
	SS	19,058	206	216.61	219.13	0.999213	4.5	0.0191	206.00	88.53
*S. sinense* Roxb. cv Uba	UL	16,004	394	419.22	429.65	0.997438	4.92	0.0228	394.00	94.76
	UR	21,903	220	223.81	224.09	0.999543	4.07	0.0389	220.00	13.87
	US	25,611	256	277.85	283.00	0.998907	3.07	0.2503	256.00	87.83

## Data Availability

The study’s datasets are available in NCBI repositories and accession numbers are presented in the article.

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
