# Peer review of "Unraveling Nitrogen Fixing Potential of Endophytic Diazotrophs of Different Saccharum Species for Sustainable Sugarcane Growth"

_ijms, 2022, doi:10.3390/ijms23116242_

Round 1

Reviewer 1 Report

The success of this phenomenon is determined by biotic and abiotic factors in the environment in which the symbiosis takes place.

Effective colonization of plants by endophytic bacteria has an impact, first of all, on the development of plants and their growth.

This publication accurately characterizes the microbial population of the roots, leaves and stems of five Saccharum species. Unfortunately, there is little information in the discussion on the practical use of these results in the future. Can the identified endophytic bacteria that have the ability to fix atmospheric nitrogen be used to create vaccines? Is it possible to reduce the amount of synthetic fertilizers thanks to this? What does the latest literature say about it?

Many authors claim that the drawback of biopreparations currently offered as vaccines with the use endophytic bacteria is the scarcity of their scientifically proven effectiveness and the attribution of excessive and unjustified effects to them, making the wider use of biopreparations controversial and often opposed by the scientific community.

Please respond to this issue in the Discussion.

Author Response

Response to Reviewer 1 Comments

Thank you very much for giving us an opportunity to revise our manuscript. We appreciate very much your constructive comments and suggestions on our manuscript entitled “Unraveling Nitrogen Fixing Potential of Uncover Novel Endophytic Diazotrophs of different Saccharum species for sustainable sugarcane growth” (Manuscript ID: ijms-1694952). Those comments are very helpful for revising and improving our paper, we have studied the comments carefully and made corrections which we hope meet with approval. The main corrections are marked in red in the revised manuscript (manuscript_891188_revision.docx).

Point 1: Unfortunately, there is little information in the discussion on the practical use of these results in the future.

Response 1: Thank you. We are grateful for this suggestion. We edited the text based on your helpful suggestions and added practical use of these results in the discussion (Page No. 17-18 & Line No: 495-503).

However, this study is mainly based on various bioinformatics tools, and based on these results analysis we found many unexplored diazotrophic bacteria from root tissues of different sugarcane species that may play an important role in sugarcane growth improvement.

Metagenomics is a revolution for the limitations of culture-based methods, which have seen a significant growth in application in recent decades. Unfortunately, due to the limits of the culture-based technique, there are still many microorganisms that cannot be identified and cultured. Therefore, in this method without a culturing method in the laboratory, DNA is extracted directly from environmental materials, and this technique can help researchers overcome these issues (Prayogo et al., 2020). This article also discusses the potential of metagenomics approach to facilitate the development of uncovering novel nifH gene producing microbes from sugarcane as yet uncultured, which helped researchers for isolate and identified that are beneficial to agriculture.

Point 2: Can the identified endophytic bacteria that have the ability to fix atmospheric nitrogen be used to create vaccines?

Response 2: Yes, the identified endophytic bacteria have the ability to fix atmospheric nitrogen and are used as biofertilizers. Various direct and indirect methods have been used to quantify the amount of nitrogen fixed by these bacteria, including the acetylene reduction assay (ARA), which is a quick but indirect method, and the 15N isotopic dilution assay, which is a robust and accurate method. In this study, we have also examined a molecular method for examining the nitrogen-fixing nifH genes in the selected strains. It has been reported that the amplification of the nifH gene is useful for confirming the potential strains showing nitrogen fixation (Zehr et al., 1996).

Previously in our laboratory, different bacterial genera such as Azotobacter, Azospirillum, Bacillus, Burkholderia, Delftia, Herbaspirillum, Pseudomonas, Enterobacter, Gluconacetobacter, Kosakonia, Stenotrophomonas, Pantoea, Klebsiella, and Serratia, etc. have been shown to be associated with biological nitrogen fixation and growth promotion in sugarcane and other crops worldwide.

Point 3: Is it possible to reduce the amount of synthetic fertilizers thanks to this? What does the latest literature say about it?

Response 3: Yes, several recent kinds of literature stating that nitrogen-fixing microbes have the potential to reduce the amount of synthetic fertilizers. Some earlier reports there are very well-known and well-studied biological nitrogen fixation microbes that have been used as biofertilizers to promote plant growth, nitrogen-fixation as well as biocontrol. Biological nitrogen fixation has been confirmed to give 30–80% of the total N for the sugarcane (Boddey et al., 1991; Döbereiner, 1997; Taulé et al., 2012; Urquiaga et al., 2012; Santi et al., 2013; Li et al., 2017; Guo et al., 2020; Singh et al., 2020; Singh et al., 2021).

Several nitrogen-fixing bacteria have been reported from inside and rhizosphere of sugarcane plants can fix nitrogen related to sugarcane plants (Gillis et al., 1989; Sevilla et al., 2001; Oliveira et al., 2002; Baldani and Baldani, 2005; Li et al., 2017; Guo et al., 2020; Singh et al., 2021; Singh et al., 2022).

Point 4: Many authors claim that the drawback of biopreparations currently offered as vaccines with the use endophytic bacteria is the scarcity of their scientifically proven effectiveness and the attribution of excessive and unjustified effects to them, making the wider use of biopreparations controversial and often opposed by the scientific community. Please respond to this issue in the Discussion

Response 4: Thank you very much for your comments. We are grateful for this suggestion. We have added it to the discussion (Page No. 17 & Line No: 479-494).

Microbial endophyte is a promising area of study in plant biology and a substantial amount of research about endophytic microbes and their positive impact on plants has been established (Babalola et al., 2007; Adedeji et al., 2020). The increased number of scientific pieces of literature over the last three decades demonstrates that scientists are more concerned with the study of endophytic microorganisms (Compant et al., 2005; Kandel et al., 2017; Jia et al., 2016). Endophytic bacterial inoculants as microbial biofertilizer alternatives would be a potential source of chemical fertilizers and have industrial applications. Plant-endophytic microbe interactions can be broadly characterized as neutral, useful (positive), or damaging (negative), while various types of connections are known to occur in ecological relationships (Schulz and Boyle, 2005). The use of root bacterial endophytes in the development of bioinoculants has been successful, and their application in current agricultural methods appears promising (Afzal et al., 2019). Numerous studies have reported endophytic bacteria that can promote the growth of plants like wheat, rice, canola, potato, tomato, sugarcane, and many more (Mei and Flinn, 2010; Sturz and Nowak, 2000; Singh et al., 2020; Singh et al., 2021; Di et al., 2022). However, endophytes' diverse host range makes them an effective tool in agricultural biotechnology. As a result, endophytes have enormous potential for application as biofertilizers and biopesticides in the development of a sustainable, safe, and efficient agricultural system.

Reviewer 2 Report

The manuscript by Singh et al. reports bacterial community composition in five varieties (species) of Saccharum with focus on diazotrophs. The experiments are well designed, and the results are of value, but in its current form the manuscript is not acceptable for publication. The primary shortcoming lies in the week presentation of results, so that this reviewer can only evaluate the discussion once the results are clear. The authors are requested to attend to the following matters and submit a revised manuscript for re-evaluation.

  1. Title: You used previously reported nifH sequences to allocate to taxa, so how can you claim that these are all novel? Novel would imply that they have not been reported to date.
  2. The term “OTU” appears as “OUT” – likely due to the autocorrect function of Word.
  3. Line 40 – 41: What is meant by “physiochemical properties of the microbial community”? You determined partial 16S sequences, so how could you infer this sort of information?
  4. Line 59: Replace “identified” with “used”.
  5. Line 71 – N2 fertilizer? Do you mean ammonia or nitrate
  6. Line 78: This is not true. For example, there are high numbers of N-fixing cyanobacteria in fresh and seawater bodies.
  7. Line 88: Not true. There have been culture-independent nifH studies conducted for over two decades – since the design of the first nifH primer sets
  8. Line 97: How do you know whether a specific 16S sequence represents an unculturable strain? The data will represent a mix of culturable and unculturable strains.
  9. Line 98: You state that unculturable organisms are the focus of this manuscript, but you present a lot of data based on isolates obtained into culture.
  10. Line 100: If this were true, the plant biotech industry would have created N-fixing plants a long time ago. Do you mean that endophytes transfer from soil to plant? Furthermore the references cited do not support these claims.
  11. Lines 108 – 110: How could omics studies improve culturing approaches? Yes, this has been shown through genome sequencing, assembly and metabolic pathway reconstruction of specific strains, but not in a banket manner.
  12. Line 128: The methods mentioned only root samples were processed for isolation. Please check to be consistent.
  13. Fig 1: The data in panel A is also reflected in panel B, just with more detail, so why include panel A?
  14. Table 1 column 2 reports “nifH gene amplification”. Without DNA sequence data this column cannot be included, for at least two reasons:
    1. All nifH primer sets known to me amplify non-specifically, so a band on a gel is not sufficient evidence for presence of the gene.
    2. The Bacillus genus, with the exception of B. nealsonii and B. caseinolyticus simply does not contain nifH genes, despite a string of papers stating such. I state this with confidence based on the recent study of all Bacillus genomes available at NCBI (Microorganisms 2021, 9(8), 1662; https://doi.org/10.3390/microorganisms9081662

Again, there are many papers that report nifH from Bacillus, but this is associated with one of two patters: Either no sequence is given, or the reported sequence aligns with that of nifH of alfa Proteobacteria. By reporting nifH for Bacillus, the results for all other isolates are in question. I am sorry to be so harsh, but the literature is already so full of questionable nifH data.

  1. Line 159: Depending on the primerset used, the nifH product is just over or under 400 bp. 200 bp is questionable. Please explain – are these partial reads from the sequencing? If so, how could you do alignment etc without loosing half the sequence?
  2. Table 2: What does the second to last column mean? Also, what does “optimized” mean – average?
  3. Lines 171 – 173: What does this mean?
  4. Line 180: “When the curve tends to be flat”. Do you mean when it reaches saturation?
  5. Line 186: DO you mean common or unique?
  6. Figure 3 reports the same data as Figure 8, the latter with clustering, so why not show it only once?
  7. Figure 2B: In the legend it speaks of “distinct and shared”, but what does shared mean in this context – between any two or the whole lot?
  8. Figure 2 C: The symbols and fonts are too small to decipher.
  9. Figure 3 legend – is this 16S or nifH? I assume 16S, but it should be stated.
  10. Figure 3 and 8: The listing of three plant genera should be explained.
  11. Figure 4: This figure conveys no information in its present form as a) the labels are far too small and low resolution to read, and b) there is no info on what A, B and C represent in the legend.
  12. 5: This strikes me as one of those analyses that are possible, but do not convey additional insights. Also, the legend states “species” when the data indicates genus level info.
  13. Figure 6: The diversity indices should best be shown as the values determined using all OUT in the sample (or then a normalized sub-set). This is best doen using a small table.
  14. How were the nifH sequences obtained allocated to taxa? The authors mention “Usearch” but this is not, at least to this reviewer’s knowledge, a suitable information source for nifH
  15. Figures 3, 8 and 9 mention plant taxa such as Sorghum and Saccharum (and 3 and 8 also Zea) when the authors claim to have used 16S- specific primers. This needs to be clarified.
  16. Figure 7: This appears to be four ways of showing the same data. As the labels are too small to read, at least in what I assume are B and C (only A is indicated), why not pick one and enlarge that with sufficient resolution?
  17. Figure 8: Label size and resolution!
  18. Figure 11: Is one more permutation of the data really necessary? What does the cladogram tell us about the microbiota in these plants?
  19. The legends to supplementary figures are too brief, with too few details – e.g. marker sizes for the gel.
  20. Discussion: No comments in this round of review.
  21. Line 611: What was “a specific primer”? What was “the target fragment”? If you used polF and polR, was this the first or second set in what is implied to be a nested PCR?
  22. Line 627 What size band was used to proceed with – especially as you report ealier a range of 400 – 200?
  23. Line 631 and following: Why was primer specificity tested, and more importantly, how? What controls were used for this?
  24. Line 667: Why mention fungal here when no fungal primers were mentioned?
  25. Conclusions: No comments in this round of review
  26. The manuscript abounds with grammatical errors which should be attended to.

Author Response

Response to Reviewer 2 Comments

Thank you very much for giving us an opportunity to revise our manuscript. We appreciate very much your constructive comments and suggestions on our manuscript entitled “Unraveling Nitrogen Fixing Potential of Uncover Novel Endophytic Diazotrophs of different Saccharum species for sustainable sugarcane growth” (Manuscript ID: ijms-1694952). Those comments are very helpful for revising and improving our paper, we have studied the comments carefully and made corrections which we hope meet with approval. The main corrections are marked in red in the revised manuscript (manuscript_891188_revision.docx).

Point 1: Title: You used previously reported nifH sequences to allocate to taxa, so how can you claim that these are all novel? Novel would imply that they have not been reported to date.

Response1: Yes, you are right. We modify the title. Here novel means we try to find uncover novel endophytic diazotrophs present in different Saccharum species.

Point 2: The term “OTU” appears as “OUT” – likely due to the autocorrect function of Word.

Response 2: Changes made as suggested.

Point 3: Line 40 – 41: What is meant by “physiochemical properties of the microbial community”? You determined partial 16S sequences, so how could you infer this sort of information?

Response 3: Thank you for your valuable comments. According to your suggestion, we have modified it as follows:

These data were assessed to ascertain the structure, diversity, abundance, and relationship between the microbial community.

Point 4: Line 59: Replace “identified” with “used”.

Response 4: Changes made as suggested.

Point 5: Line 71 – N2 fertilizer? Do you mean ammonia or nitrate.

Response 5: Added in the text (Urea).

Point 6: Line 78: This is not true. For example, there are high numbers of N-fixing cyanobacteria in fresh and seawater bodies.

Response 6: Thank you. According to your suggestion, we have modified it as follows:

Only microorganisms found in the soil, rhizosphere and plant tissues including cyanobacteria in fresh and seawater bodies are recognized to be accomplished by the N2-fixation, termed as diazotrophs.

Point 7: Line 88: Not true. There have been culture-independent nifH studies conducted for over two decades – since the design of the first nifH primer sets

Response 7: Thank you for your comments. We modified the text according to your suggestion.

Point 8: Line 97: How do you know whether a specific 16S sequence represents an unculturable strain? The data will represent a mix of culturable and unculturable strains.

Response 8: Thank you for your query.

Previously, so many reports are available for the study of unculturable methods and used as a specific 16S rRNA gene sequence in various crops. For decades, the 16S small subunit ribosomal RNA (rRNA) gene has been the gold standard marker for microbial molecular taxonomic research (Woese and Fox, 1977; Meola et al., 2015), as this highly conserved gene contains nine rapidly evolving hypervariable regions that aid in species identification (Yuan et al., 2015). Amplicon sequencing, targeting the 16S rRNA, is a high-throughput method used to study agriculture, aquatic, terrestrial, food- and host-associated microbial communities (Logares et al., 2014; Polka et al., 2015; Jiang et al., 2016; Jousselin et al., 2016; Jouglin et al., 2019; Suenami et al., 2019; Ziegler et al., 2019).

Also, the use of 16S rRNA gene sequences to study bacterial phylogeny and taxonomy has been by far the most common housekeeping genetic marker used for a number of reasons. These reasons include:

(i) its presence in almost all bacteria, often existing as a multigene family or operons;

(ii) the function of the 16S rRNA gene over time has not changed, suggesting that random sequence changes are a more accurate measure of time (evolution); and

(iii) the 16S rRNA gene (1,500 bp) is large enough for informatics purposes (Patel, 2001). 

In this research, we focused on both unculturable and culturable diazotrophic bacterial endophytes (DBEs) present in different sugarcane species.

Point 9: Line 98: You state that unculturable organisms are the focus of this manuscript, but you present a lot of data based on isolates obtained into culture.

Response 9: Thank you very much for your valuable comments.

Yes, the main focus of this study was to explore the unculturable diversity of diazotrophs. But for well understanding, we also tried to study the diversity of bacterial strains based on the culturable method.

Point 10: Line 100: If this were true, the plant biotech industry would have created N-fixing plants a long time ago. Do you mean that endophytes transfer from soil to plant? Furthermore the references cited do not support these claims.

Response 10: Thank you for your comments, Re-wrote the sentence and corrected it (Page No. 3 & Line 101-106).

Point 11: Lines 108 – 110: How could omics studies improve culturing approaches? Yes, this has been shown through genome sequencing, assembly and metabolic pathway reconstruction of specific strains, but not in a banket manner.

Response 11: Yes, you are right.

Metagenomics is a revolution against the limitations of culture-based methods, which have seen significant growth in application in recent decades. Unfortunately, due to the limits of the culture-based technique, there are still many microorganisms that cannot be identified and cultured. Therefore, in this method without a culturing method in the laboratory, DNA is extracted directly from environmental materials, and this technique can help researchers overcome these issues (Prayogo et al., 2020). This article also discusses the potential of the metagenomics approach to facilitate the development of uncovering novel nifH gene producing microbes from sugarcane as yet uncultured, which helped researchers for isolate and identified that are beneficial to the agriculture (Page No. 3 & Line 112-116).

Point 12: Line 128: The methods mentioned only root samples were processed for isolation. Please check to be consistent.

Response 12: Changes made as suggested.

Point 13: Fig 1: The data in panel A is also reflected in panel B, just with more detail, so why include panel A?

Response 13: We thank you very much again for your valuable comments. We described the detail of panel A and panel B.

(A) Total number of bacterial strains isolated from different sugarcane species in different tissues i.e., root, stem, and leaf.

(B) Number of individual isolates from different tissues (root, stem, and leaf) in the different mediums [(A) Ashby’s Glucose Agar, (B) Ashby’s Mannitol Agar, (C) Burk’s Medium, (D) Jensen’s Agar, (E) Nutrient Agar, (F) Luria-Bertani Agar, (G) Yeast Mannitol Agar (H) DF salts minimal medium, (I) Pikovskaya’s Agar and (J) Siderophore medium] of sugarcane species; (A) Saccharum officinarum L. cv Badila, (B) S. barberi Jesw.cv pansahi, (C) S. robustum, (D) S. spontaneum, and (E) S. sinense Roxb.cv. Uba).

Point 14: Table 1 column 2 reports “nifH gene amplification”. Without DNA sequence data this column cannot be included, for at least two reasons:

    1. All nifH primer sets known to me amplify non-specifically, so a band on a gel is not sufficient evidence for presence of the gene.
    2. The Bacillus genus, with the exception of B. nealsonii and B. caseinolyticus simply does not contain nifH genes, despite a string of papers stating such. I state this with confidence based on the recent study of all Bacillus genomes available at NCBI (Microorganisms 20219(8), 1662; https://doi.org/10.3390/microorganisms9081662

Response 14: Yes, you are right.

Firstly, we investigated the nifH gene for all the selected isolates, the genomic DNA was extracted and used to detect the PCR products with an accurate band size of about 360 bp according to the method of Poly et al. (2001) using the primers PolF and PolR. The amplified PCR products were determined by sequencing of PCR products, and the gene sequences were verified using the BlastN search in the NCBI GenBank database. The sequence obtained had similarity levels that varied from 95 to 100%.

After, the nitrogen-fixing ability of all isolates was tested by using the Acetylene Reduction Assay method previously described by Hardy et al. (1968).

Again, there are many papers that report nifH from Bacillus, but this is associated with one of two patters: Either no sequence is given, or the reported sequence aligns with that of nifH of alfa Proteobacteria. By reporting nifH for Bacillus, the results for all other isolates are in question. I am sorry to be so harsh, but the literature is already so full of questionable nifH data.

Response: Thank you very much for your comments and suggestions. The comments and suggestions are valuable and very helpful for revising and improving our manuscript. The manuscript is revised following your valuable suggestions. We hope that these revisions improve the quality of the manuscript.

It has been reported that the amplification of the nifH gene is useful for confirming the potential strains showing nitrogen fixation (Zehr and Capone, 1996). The lack of nifH gene amplification does not imply that the isolates are not capable of BNF since the nucleotide sequences of the nifH gene in some Bacillus species may be significantly different from others (Zehr et al., 2003).

Many Bacillus species fix N2 and the occurrence of N2 fixing bacteria in sugarcane was first reported by Dobereiner and Ruschel (1958), which was confirmed by later studies (Boddey et al., 2003; Dobereiner, 1997; Baldani et al., 2002; Singh et al., 2020). Bacillus is a bacterial genus that has been described as containing plant growth-promoting rhizobacteria (PGPR) that shifted in response to the N content (Fisher et al., 1999; Ribera et al., 2002).

Point 1: Line 159: Depending on the primer set used, the nifH product is just over or under 400 bp. 200 bp is questionable. Please explain – are these partial reads from the sequencing? If so, how could you do alignment etc without loosing half the sequence?

Response 1: Yes, you are right but I am using 360 bp. Re-wrote the sentence and corrected it.

Point 2: Table 2: What does the second to last column mean? Also, what does “optimized” mean – average?

Response 2: Yes, we check and corrected. This is the optimized average sequence length.

Point 3: Lines 171 – 173: What does this mean?

Response 3: Re-wrote the sentence and corrected it (Page No. 6 & Line No. 184-189).

Point 4: Line 180: “When the curve tends to be flat”. Do you mean when it reaches saturation?

Response 4: When the curves tended to be flat, indicating that the number of reads was significant enough to reflect the species richness.

Point 5: Line 186: DO you mean common or unique?

Response 5: Corrected; the number of unique OTUs.

Point 6: Figure 3 reports the same data as Figure 8, the latter with clustering, so why not show it only once?

Response 6: Changes made as suggested. And deleted the Figure 3.

Point 7: Figure 2B: In the legend it speaks of “distinct and shared”, but what does shared mean in this context – between any two or the whole lot?

Response 7: We corrected.

The outermost circle represents the sample name; the petals include two rows of numbers, with the number of all OTUs contained in each sample and the number of OTUs unique to each sample in the following brackets; the white circle in the center represents the core OTU quantity.

Point 8: Figure 2 C: The symbols and fonts are too small to decipher.

Response 8: Changes made as suggested.

Point 9: Figure 3 legend – is this 16S or nifH? I assume 16S, but it should be stated.

Response 9: Figure 3 is deleted as per your suggestion.

Point 10: Figure 3 and 8: The listing of three plant genera should be explained.

Response 10: Figure 3 is deleted, and Figure 8 Changes made as suggested.

Point 11: Figure 4: This figure conveys no information in its present form as a) the labels are far too small and low resolution to read, and b) there is no info on what A, B and C represent in the legend.

Response 11: Yes, you are right. We changed the figure and when we zoom the figure the label is more visible. And, we also provide the information A, B, and C in the legend.

Point 12. 5: This strikes me as one of those analyses that are possible, but do not convey additional insights. Also, the legend states “species” when the data indicates genus level info.

Response 12: Changes made as suggested.

Point 13: Figure 6: The diversity indices should best be shown as the values determined using all OUT in the sample (or then a normalized sub-set). This is best doen using a small table.

Response 13: Changes made as suggested, and deleted the figure 6.

Point 14: How were the nifH sequences obtained allocated to taxa? The authors mention “Usearch” but this is not, at least to this reviewer’s knowledge, a suitable information source for nifH

Response 14: We used the open reference OTU picking pipeline in USEARCH 64 bit v8.0.1517 (Edgar, 2010) to quality control and cluster the sequences, and a rarefying function to normalize sequencing depth across the samples.

Point 15: Figures 3, 8 and 9 mention plant taxa such as Sorghum and Saccharum (and 3 and 8 also Zea)

when the authors claim to have used 16S- specific primers. This needs to be clarified.

Response 15: Yes, you are right. But I am not using 16S specific primes. I am using a nifH gene primer.

The genome organization and expression dynamics are poorly understood in complex polyploid organisms, such as sugarcane. Therefore, the data generated can be further aligned with that of well-described model species, or of closely-related major crop species (Varshney et al., 2009), and compared with the other small genome members of the grass family (Parida, et al., 2010). The genomic similarity between sugarcane and sorghum has been frequently used to characterize the sugarcane genome (Jannoo et al., 2007; Garsmeur et al., 2011, 2018; Vilela et al., 2017; Mancini et al., 2018).

Point 16: Figure 7: This appears to be four ways of showing the same data. As the labels are too small to read, at least in what I assume are B and C (only A is indicated), why not pick one and enlarge that with sufficient resolution?

Response 16: Changes made as suggested.

Point 17: Figure 8: Label size and resolution!

Response 17: Changes made as suggested.

Point 18: Figure 11: Is one more permutation of the data really necessary? What does the cladogram tell us about the microbiota in these plants?

Response 18: Thank you for your comment.

The rings in the cladogram from inside to outside show phylum to genus diazotrophic taxonomic levels (Page No. 15 & Line No. 423-432).

Point 19: The legends to supplementary figures are too brief, with too few details – e.g. marker sizes for the gel.

Response 19: Changes made as suggested.

Point 20: Discussion: No comments in this round of review.

Response 20: Thank you. We are grateful for this suggestion. We will be happy to edit the discussion, based on your helpful comments in the next round of review.

Point 21: Line 611: What was “a specific primer”? What was “the target fragment”? If you used polF and polR, was this the first or second set in what is implied to be a nested PCR?

Response 21: Thank you for your query.

The nifH gene was amplified with primers Pol-F (TGCGAYCCSAARGCBGACTC) and Pol-R (ATSGCCATCATYTCRCCGGA), (Poly et al., 2001). The target fragment of nifH gene amplification, produces an amplified fragment of about 360 bp length.

The library was constructed by a two-step PCR amplification method. First, a specific primer (Inner primer) was used to amplify the target fragment, and the target fragment was subjected to gel recovery, and then the recovered product was used as a template for secondary PCR amplification (Outer primer) and the purpose is to add the adapter, sequencing primer, and barcode of the illumina platform to both ends of the target fragment.

According to the experimental requirements, the primers were designed as follows:

F inner primer:5'-TTCCCTACACGACGCTCTTCCGATCT-specific primer-3'

F outer primer:5'-AATGATACGGCGACCACCGAGATCTACAC- barcode - TCTTTCCCTACACGACGCTC -3'

R inner primer:5'-GAGTTCCTTGGCACCCGAGAATTCCA-specific primer-3'

R outer primer:5'-CAAGCAGAAGACGGCATACGAGAT- barcode -

GTGACTGGAGTTCCTTGGCACCCGAGA-3'

Point 22: Line 627 What size band was used to proceed with – especially as you report ealier a range of 400 – 200?

Response 22: Re-write the sentence, and mentioned the appropriate band size.

Point 23: Line 631 and following: Why was primer specificity tested, and more importantly, how? What controls were used for this?

Response 23: Thank you for your query.

NifH gene is a degenerate primer and many of primers will amplify genes that do not mediate nitrogen fixation, and thus it would be advisable for researchers to screen their sequencing results for the presence of non-target genes before analysis. This analysis will be of great utility to those engaged in molecular analysis of nifH genes from isolates and environmental samples (Gaby and Buckley, 2012). Sterile water is used as a control sample.

Therefore, in this manuscript, we have focused only nifH gene-producing microbes and the library was constructed using a two-step PCR amplification method. First, a specific primer (Inner primer) was used to amplify the target fragment, and the target fragment was subjected to gel recovery, and then the recovered product was used as a template for secondary PCR amplification (Outer primer) and the purpose is to add the adapter, sequencing primer, and barcode of the illumina platform to both ends of the target fragment.

Point 24: Line 667: Why mention fungal here when no fungal primers were mentioned?

Response 24: Deleted

Point 25: Conclusions: No comments in this round of review

Response 25: Thank you. We are grateful for this suggestion. We will be happy to edit the conclusion, based on your helpful comments in the next round of review.

Point 26: The manuscript abounds with grammatical errors which should be attended to.

Response 26: This manuscript is revised by a native English-speaking scientist.

Round 2

Reviewer 2 Report

Thank you for your revised manuscript and for detailed responses to my comments. I have only two matters that still need to be adressed:

1: Remove the words "Uncover" and "Novel" from the title.

2:  You state that the nifH sequences obtained from the isolates aligned with nifH. To help readers use this information, these culture-derived sequences  should be deposited in Genbank, and the accession numbers included in the paper - just like you have already done for the culture-independent nifH sequence data.

Author Response

Response to Reviewer 2 Comments

Thank you very much for giving us an opportunity to revise our manuscript. We appreciate very much your constructive comments and suggestions on our manuscript entitled “Unraveling Nitrogen Fixing Potential of Endophytic Diazotrophs of different Saccharum species for sustainable sugarcane growth” (Manuscript ID: ijms-1694952). Those comments are very helpful for revising and improving our paper, we have studied the comments carefully and made corrections which we hope meet with approval. The main corrections are marked in red in the revised manuscript (manuscript_891188_revision.docx).

Thank you for your revised manuscript and for detailed responses to my comments. I have only two matters that still need to be adressed:

Point 1: Remove the words "Uncover" and "Novel" from the title.

Response: Thank you for your valuable comments. According to your suggestion, we have modified the title.

Point 2:  You state that the nifH sequences obtained from the isolates aligned with nifH. To help readers use this information, these culture-derived sequences should be deposited in Genbank, and the accession numbers included in the paper - just like you have already done for the culture-independent nifH sequence data.

Response: Thank you very much for your comments and suggestions. Yes, we have all nifH gene sequences of these endophytic bacteria and some sequences are published which are mentioned in Table 1. And, other strains sequences we will submit later after finishing another manuscript writing. Thank you for your understanding.

As we complied with all reviewer’s comments, we now request that manuscript may be accepted for publication.